# Genetic analyses of the electrocardiographic QT interval and its components identify additional loci and pathways

The QT interval is an electrocardiographic measure representing the sum of ventricular depolarization and repolarization, estimated by QRS duration and JT interval, respectively. QT interval abnormalities are associated with potentially fatal ventricular arrhythmia. Using genome-wide multi-ancestry analyses (>250,000 individuals) we identify 177, 156 and 121 independent loci for QT, JT and QRS, respectively, including a male-specific X-chromosome locus. Using gene-based rare-variant methods, we identify associations with Mendelian disease genes. Enrichments are observed in established pathways for QT and JT, and previously unreported genes indicated in insulin-receptor signalling and cardiac energy metabolism. In contrast for QRS, connective tissue components and processes for cell growth and extracellular matrix interactions are significantly enriched. We demonstrate polygenic risk score associations with atrial fibrillation, conduction disease and sudden cardiac death. Prioritization of druggable genes highlight potential therapeutic targets for arrhythmia. Together, these results substantially advance our understanding of the genetic architecture of ventricular depolarization and repolarization.

The electrocardiogram (ECG) is a non-invasive tool that captures cardiac electrical activity[1]. The QT interval (QT) represents the sum of ECG measures that estimate intervals for ventricular depolarization (QRS duration; QRS) and repolarization (JT interval; JT) at an organ level (Fig. 1). The QT is used to diagnose congenital long or short QT syndromes and acquired QT-prolongation, which are associated with an increased risk for ventricular arrhythmia and sudden cardiac death (SCD)[2–5]. Susceptibility to congenital long QT syndrome (LQTS) is mediated by rare and common variation at 15 genes including *KCNQ1*, *KCNH2*, and *SCN5A*[6,7]. However, 25-30% of LQTS cases have a negative genetic test and LQTS genes do not adequately explain the heritability of the QT in the general population, or predisposition to QT-prolongation from precipitating factors such as medication[8,9].

QT and JT phenotypes are highly correlated, whereas QRS has a modest positive and a negligible negative correlation with QT and JT, respectively[10]. While at a cellular level, repolarization starts directly after depolarization of the first ventricular cardiomyocyte, at an organ level the majority of ventricular repolarization occurs during the JT interval. Therefore, investigation of the QT and its individual components has the potential to identify both shared and specific biological mechanisms for ventricular depolarization and repolarization. Genome-wide association studies (GWAS) for QT and JT have reported common variants at genes regulating cardiac ion channels, calcium-handling proteins and myocyte structure[11,12]. GWAS for QRS have highlighted genes for sodium channels, kinases and transcription factors in cardiac embryonic development[13,14]. However, a large proportion of the heritability remains unexplained (~67% for QT, ~83% for QRS), and our limited understanding of the underlying biological networks restricts the potential translational opportunities[15–17].

We have performed the largest multi-ancestry GWAS meta-analysis to date for QT, JT, and QRS in over 250,000 individuals, to discover additional candidate genes and pathways relevant to ventricular depolarization and repolarization, identify new therapeutic

e-mail: cnewtoncheh@mgh.harvard.edu; p.b.munroe@qmul.ac.uk

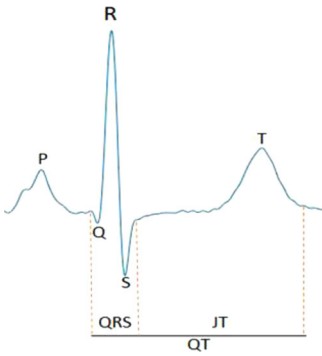

**Fig. 1 | Annotation of an example ECG signal.** QRS duration and the JT interval approximate the time periods for ventricular depolarization and repolarization on the surface ECG. The entire segment from onset of the Q wave to end of the T wave is the QT interval.

targets, and test the association of polygenic risk scores (PRSs) with cardiovascular disease.

## Results

### Meta-analysis of GWAS
Thirty-five studies contributed to our primary multi-ancestry GWAS meta-analysis for QT, JT, and QRS, comprising a maximum total of 252,977 individuals of European (84%), Hispanic/Latino (7.7%), African (6.7%), South and South-East Asian (<1%) ancestries (Supplementary Data 1–3, Supplementary Note 1). The meta-analysis workflow is summarized in Fig. 2. No evidence of inflation of test statistics was observed (Supplementary Figs. 1 and 2).

### QT GWAS meta-analysis
We discovered 176 genome-wide significant (GWS; $P$-value ($P$) $< 5 \times 10^{-8}$) lead variants at independent autosomal loci (114 unreported) associated with QT in the multi-ancestry meta-analysis (Table 1, Supplementary Fig. 3a). Of the previously reported loci for QT or JT (grouped as the phenotypic correlation is high), there was support for association at 62/66 (93.9%) loci ($P < 5 \times 10^{-8}$). There was weaker support for association at 3 loci (*NRAP*, *MYH6*, *NACA*, $P < 1.29 \times 10^{-4}$) and no evidence of support for *SUCLA2* ($P > 0.05$) (Supplementary Data 4). Ancestry-specific analyses identified additional unreported loci in European (11), Hispanic/Latino (1), and African (1) individuals (Supplementary Data 5). All European and African ancestry-specific lead variants were supported in the multi-ancestry analysis ($P < 5 \times 10^{-5}$). The Hispanic ancestry locus was not supported in the multi-ancestry analysis ($P = 0.07$), however, the lead variant at this locus was rare (MAF = 0.002) and monomorphic in Europeans.

To identify additional signals, we performed joint and conditional analyses with Genome-wide Complex Trait Analysis (GCTA)[18] using summary statistics from the European ancestry meta-analysis with the reference sample from UK Biobank (52,230 individuals of European ancestry). These analyses identified an additional 65 conditionally independent variants at 38 loci at $P_{joint} < 5 \times 10^{-8}$ (Supplementary Data 6).

### JT and QRS GWAS meta-analyses
For JT and QRS, we identified 155 and 121 lead variants at independent autosomal loci (96 and 77 unreported) in multi-ancestry meta-analyses respectively (Table 1, Supplementary Fig. 3, Supplementary Data 7 and 8). Ancestry-specific analyses identified additional unreported loci ($N = 18$) in European (4 JT, 6 QRS), African (4 JT, 2 QRS) and Hispanic/Latino (1 JT, 1 QRS) ancestries. Of these, 13 lead variants had evidence for support in the multi-ancestry analysis ($P < 5 \times 10^{-5}$). Joint and conditional analyses identified an additional 56 and 29 conditionally independent variants at 32 JT and 18 QRS loci, respectively (Supplementary Data 6).

### X-chromosome meta-analyses
X-chromosome analyses (multi-ancestry sample size: 86,600 and 60,343 for separate female and male analyses, respectively) identified one locus in males in both multi- and European ancestry meta-analyses, for QT and JT (Table 1, Supplementary Data 5 and 7). There were no GWS findings for QRS or female X-chromosome analyses, and no suggestive evidence of association on a lookup of the lead QT/JT variant (rs55891214) in these analyses ($P > 0.05$). rs55891214 is highly correlated ($r^2 > 0.9$) with lead variants reported in GWAS for serum testosterone, estradiol levels, male-pattern baldness, and heel bone mineral density[19–22]. The nearest gene, *FAM9B*, is exclusively expressed in the testis, and together these findings suggest the association may be driven by serum testosterone levels[19,23].

### Overlap of genetic contributions and heritability of QT, JT, and QRS
There was substantial overlap of multi-ancestry JT and QT GWAS loci (130/200, 65%) but less between QRS and QT (53/243, 21.8%) (overlap: $r^2 > 0.1$ between lead variants or within ±500 kb). For QRS and JT, there was overlap at 34 loci, where a lead variant was genome-wide significant in both analyses (Supplementary Data 9). Predominantly discordant (27/34, 79.4%) directions of effect were observed at these lead variants (Fig. 3). Across all loci for QT, JT and QRS, overlap was observed with previously reported loci for PR interval (51 (29.1%), 46 (29.7%), 42 (34.7%), respectively) and resting heart rate (58 (33.1%), 57 (36.8%), 46 (38.0%)), demonstrating shared genetic contributions. 13 loci were common to all 5 ECG measures (Supplementary Data 10), highlighting these loci as integral genetic determinants of global cardiac electrophysiology. Estimated genetic correlations were calculated using LD Score Regression (LDSC)[24,25]. A strong positive correlation was observed between QT and JT ($r_g = 0.91$, $P < 0.001$) and a weak positive correlation between QT and QRS ($r_g = 0.17$, $P = 0.05$) (Supplementary Fig. 4). In contrast, a negative genetic correlation was observed between JT and QRS ($r_g = -0.25$, $P = 0.003$).

SNP-based heritability estimations in Europeans from UKB for QT, JT and QRS were 29.3%, 29.5% and 15.0% (standard error [SE]:1%) respectively. The percentage of overall variance explained by all lead and conditionally independent variants from the European meta-analysis was 14.6%, 15.9% and 6.3%. Therefore, these variants explain 49.8%, 53.9%, and 42.0% of the SNP-based heritability of QT, JT and QRS in the UKB individuals included in the heritability calculations (Supplementary Note 2).

### Gene-based meta-analysis
To investigate whether rare variants (MAF < 0.01) in aggregate modulate ECG traits, we conducted gene-based meta-analyses of rare variants predicted by Variant Effect Predictor (VEP)[26] to have high or moderate impact on protein function, using Sequence Kernel Association Testing (SKAT)[27]. These analyses discovered 13, 16, and 3 genes for QT, JT and QRS, respectively ($P < 2.5 \times 10^{-6}$; Bonferroni adjusted for ~20,000 genes). These genes were brought forward for conditional analyses, and 7, 7 and 2 genes remained associated with QT, JT and QRS, respectively, after conditioning on the rare variant with the lowest $P$-value at each gene ($P < 0.05$/number of genes) (Table 2). These results indicate that the gene-based associations were not a consequence of a single variant with a strong effect. We identified an association of rare variants in aggregate at Mendelian long-QT syndrome (LQTS) genes (*KCNQ1* [QT and JT], *KCNH2* [QT]). *SCN5A* was associated with JT and QRS, however, it did not reach the Bonferroni corrected threshold for significance for QT. This could be explained by the discordant directions of effect observed for QRS (Beta [β]: 0.04) and JT (β:−0.05), which subsequently reduce the strength of the association observed for QT (β:−0.03). *MYH7* and *TNNI3K* were also associated with JT. *TNNI3K*, which was not associated using single variant analysis, encodes a cardiomyocyte-specific kinase previously

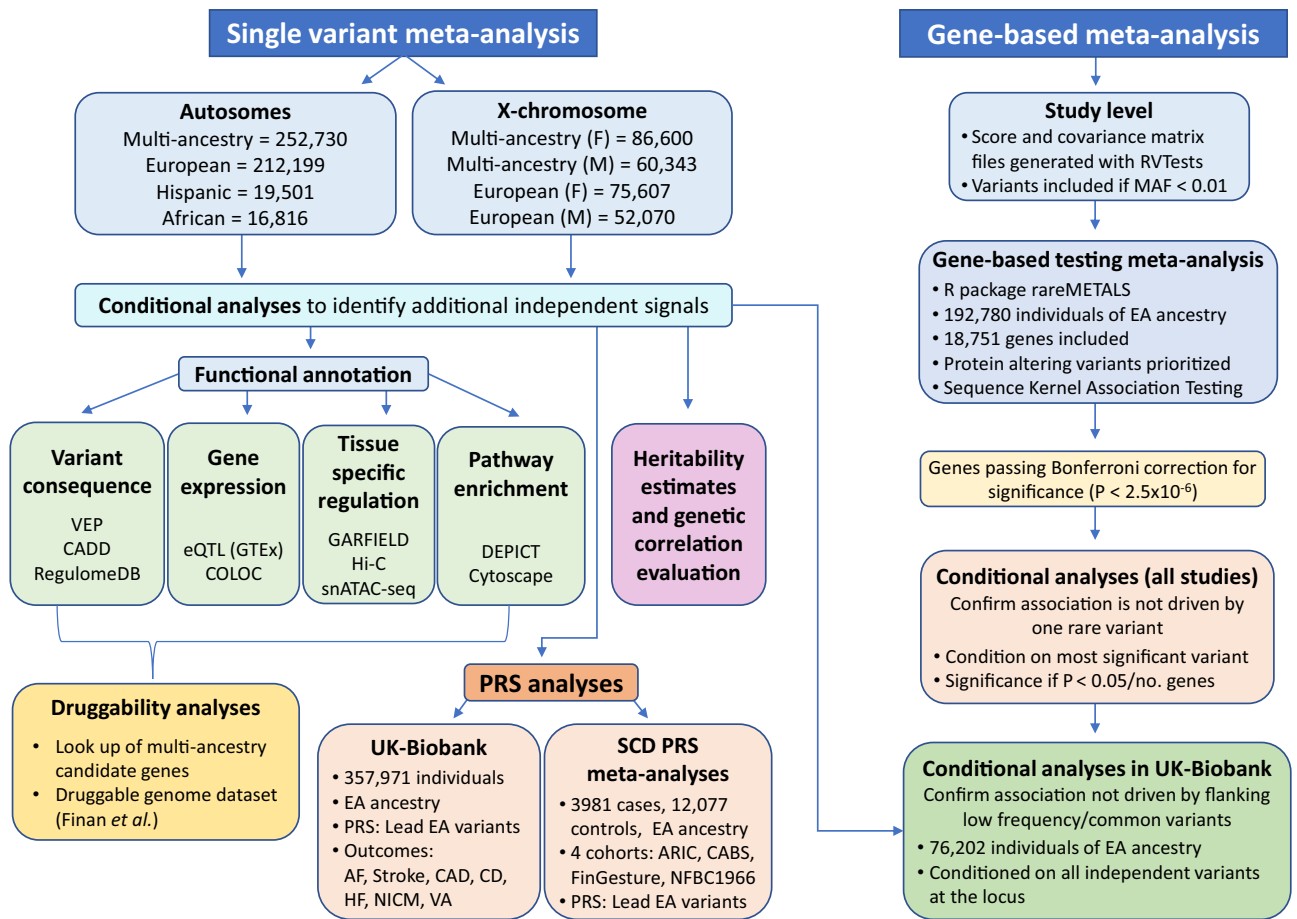

**Fig. 2 | Workflow of the genetic analyses performed for QT, JT, and QRS.** Workflow including single variant and gene-based meta-analyses, and down-stream bioinformatics. VEP (Variant Effect Predictor), CADD (Combined Annotation Dependent Depletion), eQTL (expression Quantitative Trait Locus), GTEx (Genotype-Tissue Expression project), COLOC (Colocalization), GAR-FIELD (GWAS Analysis of Regulatory and Functional Information Enrichment with LD correction), DEPICT (Data-driven Expression-Prioritized Integration for Complex Traits, GWAS (Genome-Wide Association Study), EA (European Ancestry), PRS (Polygenic Risk Score), AF (Atrial Fibrillation), CAD (Coronary Artery Disease), CD (Conduction Disease), HF (Heart Failure), NICM (Non-Ischemic Cardiomyopathy), VA (Ventricular Arrhythmia), SCD (Sudden Cardiac Death).

linked to familial cardiac arrhythmia and dilated cardiomyopathy (OMIM:613932)[28,29].

As several genes mapped to loci implicated in single variant GWAS analyses, we explored the relationship between these rare gene-based signals and common (MAF > 0.05) or low frequency (0.01 ≤ MAF ≤ 0.05) variants. Analyses were repeated in 76,202 individuals from UKB

## Table 1 | Number of loci identified in each QT, JT and QRS meta-analysis

| Autosome | Sample size | QT No. of loci | JT No. of loci | QRS No. of loci |
|---|---|---|---|---|
| Multi-ancestry | up to 252,977 | 176 | 155 | 121 |
| European | up to 212,199 | 171 | 150 | 110 |
| Hispanic | 19,501 | 13 | 13 | 13 |
| African | 16,816 | 7 | 10 | 5 |
| **X-chromosome (Males)** | **Sample size** | **No. of loci** | **No. of loci** | **No. of loci** |
| Multi-ancestry | 60,343 | 1 | 1 | 0 |
| European | 51,386 | 1 | 1 | 0 |

Number of loci identified for each ECG trait split into autosome meta-analyses and X-chromosome sex-stratified analyses.
*Sample size* maximum sample size in the meta-analysis, *No.* number.

conditioning on independent significant variants identified in the corresponding European GWAS meta-analysis, and residing within the same locus as the gene. These conditional analyses showed that associations for *KCNQ1*, *KCNH2*, and *RNF207* with QT and/or JT were independent of flanking variants identified by GWAS (Supplementary Data 11). Because conditional analyses required a large sample with a shared set of variants, we lacked adequate power to definitively determine the independence of *DLEC1* from nearby common variants. For *MYH7* and *TNNI3K*, there were no GWS variants at the locus in our single-variant meta-analysis.

**Variant-level functional annotation**
Most multi-ancestry QT lead variants, 160/176 (90.9%), were common, 13 were low frequency and 3 were rare (*RUFY1*, *DNAJB5*, *CACNB2*; Supplementary Data 5). At 25 loci, a lead variant or proxy ($r^2 > 0.8$) was annotated with VEP, as missense ($N = 24$) or stop-gain ($N = 1$) (Supplementary Data 12a). Ten of these (40%) were predicted by SIFT or PolyPhen-2 to be "deleterious" or "damaging". These included: rs1805128, a *KCNE1* polymorphism D85N (c.253G>A), which is a recognized modifier of Long QT syndrome[30], and a missense variant in *NEXN* (rs1166698). *NEXN* mutations are associated with cardiomyo-pathies (OMIM:613121)[31,32]. At 16 loci, a lead variant or proxy had a Combined Annotation Dependent Depletion (CADD) score ≥ 20 and therefore was predicted to be among the top 1% most deleterious in

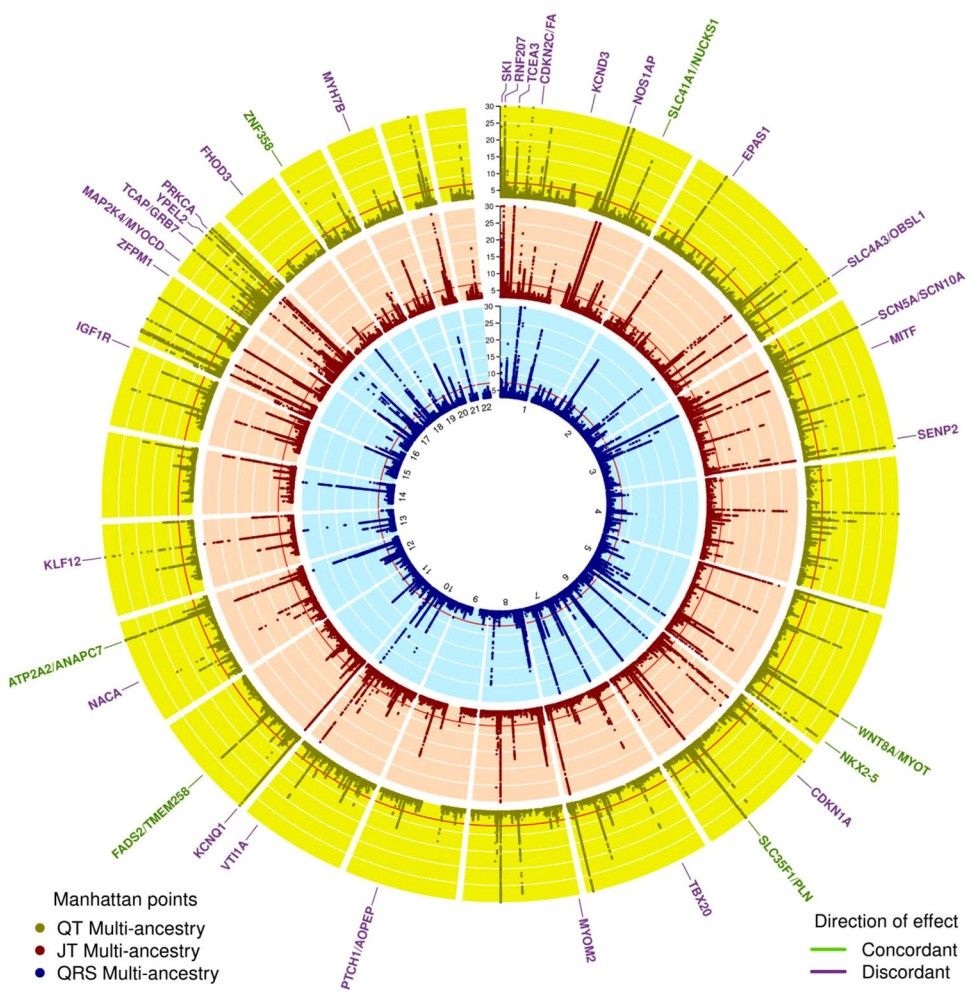

**Fig. 3 | Circular Manhattan plot for QT, JT, and QRS multi-ancestry meta-analyses.** Circular Manhattan plots for QT (outer, yellow), JT (middle, red), and QRS (inner, blue) multi-ancestry GWAS linear regression meta-analyses. The *Y*-axis has been restricted to -log10 *P*-value < 30. Two-sided *P*-values are reported. A Bonferroni-corrected threshold (<5 × 10⁻⁸) was used to declare significance. Overlapping JT and QRS loci are labeled with the most likely candidate at the locus color coded according to a concordant (green) or discordant (purple) direction of effect at a variant level. Direction of effect was compared by comparing the lead JT variant beta with the corresponding direction of effect of the same variant in the QRS GWAS meta-analysis. This plot was produced using the R package Circlize version 0.4.10. Gu, Z. (2014) circlize implements and enhances circular visualization in R. Bioinformatics. 10.1093/bioinformatics/btu393.

the genome (Supplementary Data 12b). Of these, the variants at 10 loci were non-coding.

A similar proportion of lead variants for JT (91%) and QRS (95.9%) were common. Missense variants were identified in 21 JT and 11 QRS loci (Supplementary Data 12a) and predicted to be "deleterious" or "damaging" at 8 JT and 7 QRS loci. At 18 JT and 13 QRS loci, the lead/proxy variant had a CADD score ≥ 20 (Supplementary Data 12b). Of these, the variant was non-coding in 13 JT and 7 QRS loci.

**Association with gene expression levels in cardiac tissue**
Using data from the Genotype-Tissue Expression (GTEx) project[33], we identified 39 (22.2%) multi-ancestry QT loci where a variant was a cis-eQTL in left ventricle (LV) or right atrial appendage (RAA) tissue samples (Supplementary Data 13). There was strong support for pairwise colocalization (posterior probability>0.75) at 17 loci for LV tissue and 14 for RAA.

At 37 (23.9%) JT and 18 (14.9%) QRS multi-ancestry loci the lead variant or proxy was a significant cis-eQTL in LV or RAA tissue (Supplementary Data 13). There was support for colocalization at 12 JT and 7 QRS loci in LV tissue and 12 JT and 8 QRS loci in RAA. Comparing QT/JT and QRS, discordant directions of effect were identified at 3 overlapping loci (*KLF12* [RAA], *PRKCA,* and *TCEA3* [RAA and LV]), suggesting

differences at a variant level may translate to opposing effects on tissue-specific gene expression (Fig. 4).

**Tissue- and cell-type specific effects of variants through regulatory elements**
Potential target genes of regulatory variants were identified using two long-range chromatin interaction datasets (40 kb-resolution Hi-C and ~4 kb-resolution promoter-capture Hi-C) from LV and RV tissue[34,35]. Promoter interactions were identified at 39 (22.2%) QT loci (Supplementary Data 14a and 14b). Evaluation of cardiac cell-type specific effects using single nucleus Assay for Transposase-Accessible Chromatin using sequencing (snATAC-seq) data[36], identified significant enrichment at open chromatin regions for QT in atrial and ventricular cardiomyocytes (Supplementary Fig. 5).

Promoter interactions were identified at 46 (29.7%) JT and 28 (23.1%) QRS loci (Supplementary Data 14a and 14b). Cardiac cell-type specific enrichment was significant in atrial and ventricular cardiomyocytes for both JT and QRS, and in adipocytes for JT (Supplementary Fig. 5).

Tissue-specific enrichment of variants in DNaseI hypersensitivity sites, using GWAS Analysis of Regulatory and Functional Information Enrichment with LD correction (GARFIELD, v2)[37] identified strongest

**Table 2 | Significant genes from gene-based meta-analysis for each ECG trait following conditional analysis**

|  | Gene | N | P-value Unconditional | No. of variants | Beta | SD | Conditioned variant | P-value after conditioning |
|---|---|---|---|---|---|---|---|---|
| QT | KCNH2 | 180,961 | 9.11E−12 | 43 | 0.15 | 0.022 | 7:150654525-G-A | 1.68E−07 |
|  | NOS3 | 183,747 | 1.04E−10 | 68 | 0.03 | 0.012 | 7:150698349-G-A | 1.65E−05 |
|  | KCNQ1 | 158,377 | 1.23E−11 | 36 | 0.06 | 0.047 | 11:2790111-C-T | 1.75E−05 |
|  | RNF207 | 168,015 | 9.46E−16 | 39 | −0.15 | 0.020 | 1:6279316-C-T | 3.89E−05 |
|  | OLFML2B | 183,747 | 5.30E−12 | 46 | 0.06 | 0.012 | 1:161970046-A-G | 6.85E−04 |
|  | TSSC4 | 189,264 | 5.16E−18 | 25 | −0.10 | 0.014 | 11:2424684-A-C | 1.97E−03 |
|  | MYH7 | 173,501 | 3.13E−07 | 38 | 0.05 | 0.025 | 14:23892910-A-G | 3.58E−03 |
| JT | SCN5A | 181,936 | 2.53E−15 | 90 | −0.05 | 0.013 | 3:38591853-A-G | 1.03E−05 |
|  | NOS3 | 183,468 | 5.13E−12 | 68 | 0.04 | 0.012 | 7:150698349-G-A | 1.60E−05 |
|  | RNF207 | 167,737 | 1.33E−16 | 39 | −0.16 | 0.020 | 1:6279316-C-T | 5.55E−05 |
|  | MYH7 | 173,223 | 1.48E−08 | 38 | 0.06 | 0.025 | 14:23892910-A-G | 1.47E−04 |
|  | KCNQ1 | 158,099 | 5.01E−11 | 36 | 0.10 | 0.047 | 11:2790111-C-T | 3.09E−04 |
|  | TNNI3K* | 181,936 | 3.48E−08 | 56 | 0.03 | 0.016 | 1:74929170-T-C | 2.79E−03 |
|  | OLFML2B | 183,468 | 1.35E−12 | 46 | 0.07 | 0.012 | 1:161970046-A-G | 2.91E−03 |
| QRS | SCN5A | 181,930 | 3.34E−11 | 90 | 0.04 | 0.013 | 3:38591853-A-G | 7.89E−06 |
|  | DLEC1 | 188,621 | 1.05E−06 | 117 | 0.05 | 0.012 | 3:38163900-C-T | 2.91E−04 |

N total sample size at gene-based meta-analysis for each gene, P-value unconditional Gene-based P-value (P) for association with the ECG trait as output from Sequence Kernel Association Testing in rareMETALS (see "Methods" for more information). Findings were reported as statistically significant if $P < 2.5 \times 10^{-6}$ (Bonferroni-corrected for ~20,000 genes tested), No. of variants Number of rare variants included in burden testing, SD Standard Deviation, Conditioned variant Variant with the smallest P-value used for conditional analysis to test whether the association was driven by a single variant, P-value after conditioning Gene-based P-value (as output from Sequence Kernel Association Testing in rareMETALS, after conditioning on the variant with the smallest P-value. A Bonferroni-corrected threshold (0.05/number of genes brought forward for conditional analysis for each ECG trait) was used to declare significance. Statistical tests were two-sided. *Gene within a locus previously unreported for these ECG measures using single-variant GWAS methods.

enrichment in fetal heart tissue for QT, JT, and QRS (Supplementary Fig. 6).

### Gene-set tissue/cell-type enrichment and pathway analyses

Candidate genes were prioritized to common functional pathways using reconstituted gene sets in Data-driven Expression-Prioritization Integration for Complex Traits (DEPICT) software[38]. These gene sets were highly expressed (false discovery rate [FDR] < 0.01) in cardiac tissues for all three traits (Supplementary Data 15). Additionally, for QRS, connective tissue cell types including heart valve, chondrocytes, and joint/skeletal tissues, were significant (Supplementary Fig. 7).

The most significant (FDR < 0.01) gene-ontology (GO) biological processes for QT could be grouped into broad categories, including cardiac and muscle cell differentiation/development, and regulation of gene expression (Supplementary Fig. 8). In addition, and previously not reported for QT, response to insulin and insulin receptor signaling processes were identified (Fig. 5). In the Reactome database, top pathways were related to signal transduction, including protein kinase B mediated events, mechanistic target of rapamycin (mTOR) signaling, the phosphoinositide 3-kinases cascade and the insulin receptor signaling cascade (Supplementary Data 16). The top 10 mouse phenotypes enriched for these gene sets included abnormal myocardial layer/trabeculae morphology, decreased embryo size/growth retardation, and increased heart weight.

GO biological processes significant for JT (and QT), but not QRS, included regulation of gene expression, histone and chromatin modification, insulin receptor signaling and response to insulin stimulus (Fig. 5, Supplementary Data 16). Cellular growth, the transmembrane receptor protein serine/threonine kinase signaling pathway and vasculogenesis were enriched only for QRS. Reactome pathways significant for JT (and QT) but not QRS included regulation of lipid metabolism by peroxisome proliferator and its activated receptor effect on gene expression, along with insulin receptor signaling and related events (Supplementary Fig. 9). The top Reactome pathway for QRS, not significant for JT (or QT), was extracellular matrix interactions[39]. A summary of the findings for all bioinformatic analyses for previously unreported loci is in Supplementary Data 17–19.

### Identification of potential drug targets for therapeutic opportunities

To identify potential drug targets for arrhythmia, we interrogated the druggable gene-set database published by Finan et al.[40]. We examined all 200, 173 and 155 plausible candidate genes from the QT, JT and QRS multi-ancestry meta-analyses respectively, including known and previously unreported loci. 53 (QT), 46 (JT), and 31 (QRS) genes were identified as potential drug targets, that are not current targets of anti-arrhythmic drugs (Supplementary Data 20). Of these, 21 QT, 17 JT, and 10 QRS genes were classed as Tier 1, encoding proteins that are targets of drugs either approved or in development. Genes from significant signals in cardiac tissue-specific eQTL co-localization and Hi-C analyses may be favored for prioritization. Of the 53 potential gene therapeutic targets for QT, ABCC8, KCNA7, KCNK13, PRKCA and THRB had support for co-localization in eQTL analyses and NFKB1, MITF, PLK2 and CASQ2 were significant Hi-C findings. CASQ2 and PLK2 were previously investigated as potential targets of gene transfer technology[41,42].

### Association of genetically determined QT, JT, and QRS with cardiovascular disease and sudden cardiac death

PRSs were constructed using European-ancestry lead variants to determine the relationship of genetically determined QT, JT and QRS with the directly measured ECG phenotype, and cardiovascular diseases that may have shared genetic contributions to risk. Each PRS was tested for association with the directly measured ECG trait in 4214 individuals from UKB not included in the GWAS. Associations observed for each PRS were (β [95% CI]): 6.4 ms (5.7–7.1) for QT; 6.4 ms (5.7–7.1) for JT; and 2.2 ms (1.7–2.7) for QRS (ms per standard deviation [SD] increase in the PRS). A significant difference in means was observed (two sample t-test, $P < 2.2 \times 10^{-16}$), when comparing individuals in the top and bottom quintiles of the PRS distribution (16.0 ms for QT; 16.0 ms for JT and 6.2 ms for QRS).

In ~357 K unrelated individuals of European ancestry from UKB not included in the GWAS meta-analysis, each PRS was tested for association with prevalent cardiovascular disease cases including atrial fibrillation (AF), "atrioventricular block (AVB), or permanent pacemaker implantation (PPM)", "bundle branch block (BBB) or fascicular

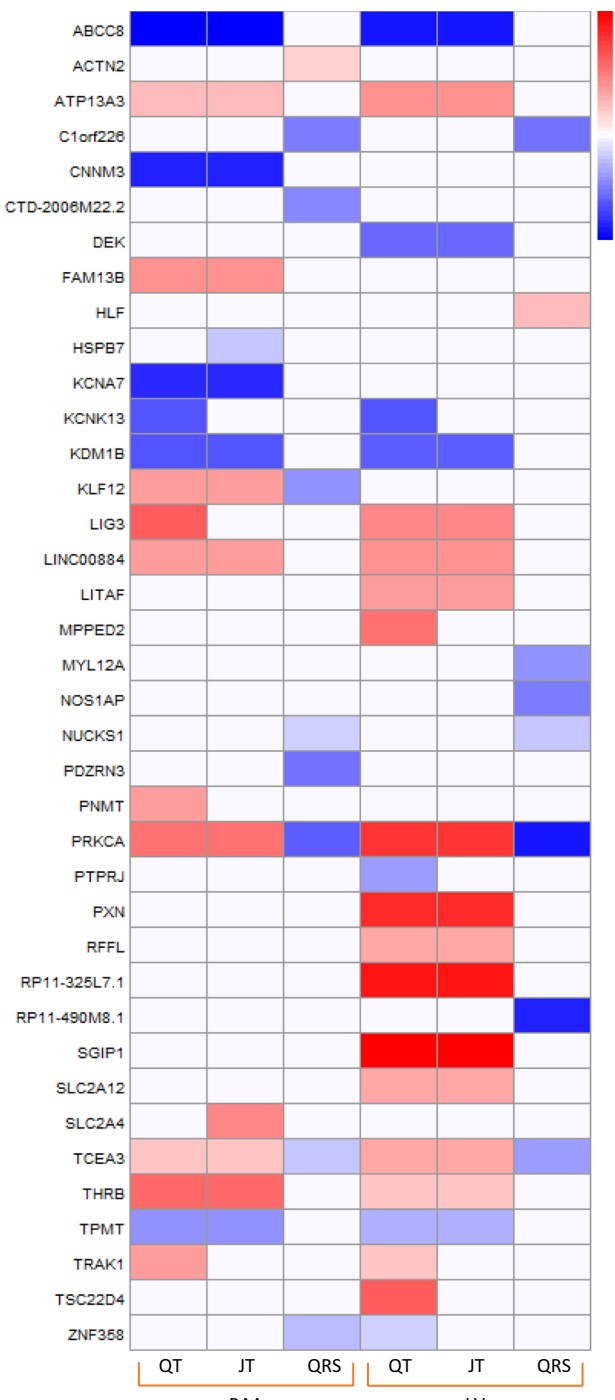

**Fig. 4 | Comparison of co-localized eQTL signals for QT, JT, and QRS in right atrial appendage and left ventricle tissues.** Colocalization analyses performed using data from GTEx (version 8), using the R package COLOC (methods). A posterior probability of >75% was used to declare significance. Boxes are color coded to show either increased (red) or decreased (blue) effect on tissue-specific gene expression. The degree of shading reflects the normalized effect sizes (and therefore no units) from the slope of the linear regression model for the effect allele relative to the non-effect allele (see methods for more information). The direction of effect has been aligned to the ECG trait prolonging allele. Y axis: Transcripts. RAA: Right atrial appendage, LV: Left ventricle.

block", and heart failure (Supplementary Data 21, Fig. 6). A Bonferroni threshold (0.05/number of conditions tested) was used to indicate significance ($P < 6.3 \times 10^{-3}$). Genetically determined QT and JT were associated with decreased risk for "AVB or PPM implantation" (odds

ratio [OR] (95% CI) per SD: 0.94 [0.924–0.963] and 0.94 [0.918–0.956] respectively). In contrast, genetically determined QRS was associated with increased risk for "BBB or fascicular block" (1.07 [1.037–1.105]). Genetically determined QT and QRS were associated with decreased risk for AF (0.97 [0.954–0.979]) and 0.93 [0.921–0.945], respectively). Including the QRS PRS as a covariate in the QT model did not substantially change the point estimate (OR: 0.97 [0.958–0.983]), indicating the relationship with AF was not driven by overlap with the genetic contribution for QRS.

As these ECG measures are established risk markers for malignant ventricular arrhythmia and SCD, we also tested each PRS for association with SCD in the Atherosclerosis Risk in Communities (ARIC) study, Cardiac Arrest Blood Study (CABS), Finnish Genetic Study for Arrhythmic Events (FinGesture) and Northern Finland Birth Cohort of 1966 (NFBC1966) cohorts (Supplementary Data 22). Results are reported as risk for SCD, per unit increase in the average ms per allele. The PRS distribution (mean [SD]) was: 0.321 (0.02) for QT; 0.361 (0.02) for JT; 0.134 (0.008) for QRS. Therefore, findings are reported as log OR (95% CI). The lead variant at the *NOS1AP* locus for QT and JT (rs12042862, T-allele) was associated with increased risk for SCD (0.11 [0.036–0.190], $P = 0.004$), as previously reported[43]. There was no association observed between each PRS and SCD in the full sample (Supplementary Data 23). As the incidence of SCD is different between men and women, we performed sex-stratified analyses[44]. Increasing QT PRS was associated with SCD in females (8.2 [3.05-13.35], $P = 1.8 \times 10^{-3}$) with a concordant direction of effect across all studies. Sensitivity analyses in FinGesture suggested the association between the QT PRS and SCD in women, may be driven by non-ischemic aetiologies compared with ischaemic ($P = 0.004$ vs. 0.926 respectively).

## Discussion

Our large-scale GWAS meta-analyses for QT and its components, QRS and JT, substantially advance our understanding of the genetic architecture of ventricular depolarization and repolarization. We more than double the number of autosomal loci associated with each trait and identify sex-specific effects at an X-chromosome locus (*FAM9B*). In addition to established processes, we report loci involved in energy metabolism and response to insulin, which have greater enrichment for ventricular repolarization. Extracellular matrix interactions, cell growth, and connective tissue components are significantly enriched among QRS-associated genes. We identify Mendelian genes for which a burden of rare variants are associated with these ECG measures (e.g. *KCNQ1, KCNH2, TNNI3K*). We also highlight potential therapeutic targets and together with the association of PRSs with AF, conduction disease and SCD, these indicate possible translational opportunities of our findings.

Previous knowledge of ventricular repolarization has centered on the role of cardiac ion channels, predominantly from the investigation of inherited arrhythmic syndromes[3]. However, ventricular repolarization is complex and influenced by multiple processes, as suggested by previous GWAS, and now advanced in our present study[11,12]. Our analyses have identified additional candidate genes involved in cardiomyocyte differentiation, tissue development, cardiac contraction and regulation of gene expression. In this study, we also report pathways related to insulin receptor and mTOR signaling, along with genes that implicate cardiac energy metabolism. Ion homeostasis is an energy-consuming process and mismatch in the supply and utilization of adenosine triphosphate (ATP) can lead to electrical and mechanical instability[45]. Appropriate cardiac energy utilization is therefore, necessary to maintain normal physiological activity. Candidate genes common to QT and JT within insulin related pathways include *SLC2A4*, *PIGQ*, and *ABCC8*. *SLC2A4* (alias *GLUT4*), is a glucose transporter in cardiomyocytes with effects on cardiac contractility, development of hypertrophy, and susceptibility to atrial and ventricular arrhythmias[46–48]. *PIGQ* is involved in the

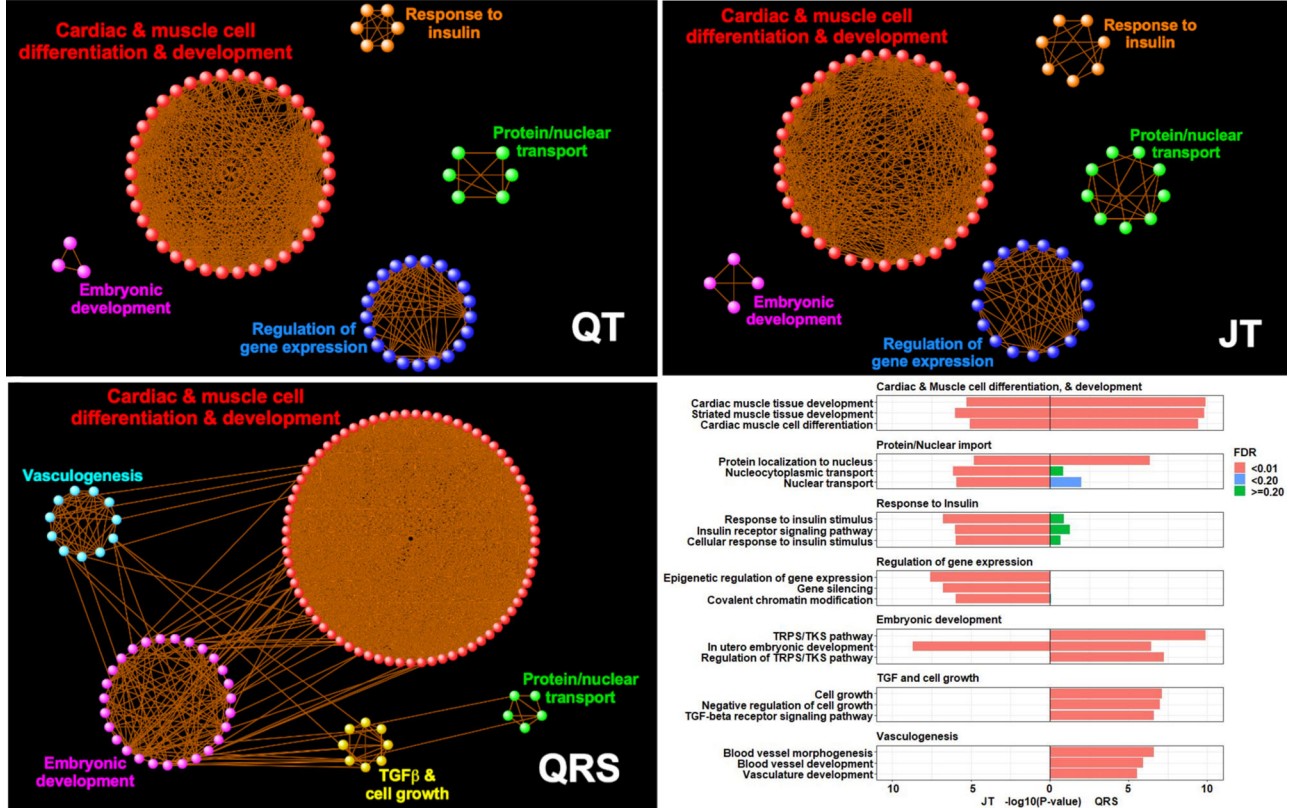

**Fig. 5 | Enrichment network visualization of DEPICT GO biological processes.** The first three panels (QT, JT, and QRS) were created using Cytoscape (v3.8.2). Significant GO biological processes (false discovery rate [FDR] < 0.01) from DEPICT pathway analyses (represented as a colored point in the image) were linked together (light orange line) when containing a minimum of 25% overlap of gene members. Orphan pathways or those with less than three edges were excluded. This created discrete "modules" of interlinked pathways, from which common themes could be identified. The final panel shows a bar graph with the most significant GO process members (Y-axis) for JT and QRS from each "common theme", along with their enrichment *P*-values (X-axis) and color coded by FDR (see legend). Enrichment *P*-values are as output by DEPICT which compares z-scores derived from Welch's *t*-test again the null hypothesis (see methods for more information). TGF-beta: Transforming growth factor beta, TRPS/TKS: transmembrane receptor protein serine/threonine kinase.

biosynthesis of glycosylphosphatidylinositol-anchored proteins, which are involved in membrane protein transportation and cell surface protection[49]. Mutations involving these proteins cause cardiac-related glycosylation disorders, including congenital defects and arrhythmia[50]. *ABCC8* (alias sulfonylurea receptor 1) modulates ATP-sensitive potassium channels and insulin release. It is a crucial component of sarcolemma $K^+$ ATP channels in mouse atrial myocytes, however a similar role has not been identified in human cardiomyocytes[51,52]. In humans, the role of *ABCC8* is predominantly within pancreatic beta cells, and therefore may indicate indirect effects on ventricular repolarization through insulin secretion[53]. Insulin is considered cardioprotective and receptors for insulin signaling are highly expressed in the heart[54]. Type-1 diabetic mice have altered ion channel kinetics in ventricular and atrial myocytes, that increase the risk of arrhythmia and can be reversed with insulin therapy[55–57].

Sex differences in ventricular repolarization are well recognized and incorporated into clinical definitions for QT-prolongation[58]. Our study identified an X-chromosome locus (*FAM9B*), which may contribute to these differences through serum testosterone levels. In rat cardiomyocytes, testosterone upregulates *KCNQ1* expression with a long-term effect on QT-shortening[59]. Androgen receptors are expressed in the atrial and ventricular myocardium in multiple species including humans of both sexes[60,61]. During puberty in males, appropriate shortening is driven by increasing testosterone, while the comparatively gradual development of a relatively hypogonadal state in post-pubertal males, may explain senescent increases[58,62]. Prolonged ventricular repolarisation is also a feature of several human diseases

that share androgen deficiency as a common characteristic[63–66]. In addition to testosterone, a role for other hormones in ventricular repolarization is supported by the association of variants with QT and JT at the *THRB* locus, a nuclear hormone receptor for triiodothyronine[67].

Loci for QRS in comparison, have greater enrichment in processes for vasculogenesis, cell growth, and embryonic development (Fig. 5). These include candidate genes encoding transcription factors (or their regulators) with roles in cellular proliferation and cardiac conduction system development (*ID2, PRDM6* and *PALLD*)[68–71]. Gene sets were also enriched in connective tissues and cell types. *PDE1A* is an example of one of these genes. It encodes a cyclic nucleotide phosphodiesterase and regulates cardiac fibroblast activation and fibrosis formation[72]. Myocardial fibrosis is a pathophysiological process in ventricular remodeling, which impairs cardiac electrical conduction and increases the risk for arrhythmogenesis[39,73].

In this study, we observed predominantly discordant directions of effect comparing overlapping QRS and JT loci. Despite the strong phenotypic and genetic correlation between QT and JT, the genetic contribution to the QT interval represents the combined effects of variants associated with QRS and JT. Therefore, overlap and shared biology are also observed across QT and QRS loci, along with a weak positive genetic correlation. These findings could inform drug development for arrhythmia, as genes or their encoded proteins could be targeted for their specific effects on predominantly ventricular depolarization or repolarization.

Inherited arrhythmic syndromes highlight the importance of rare variation on ECG traits and arrhythmic risk. However, our

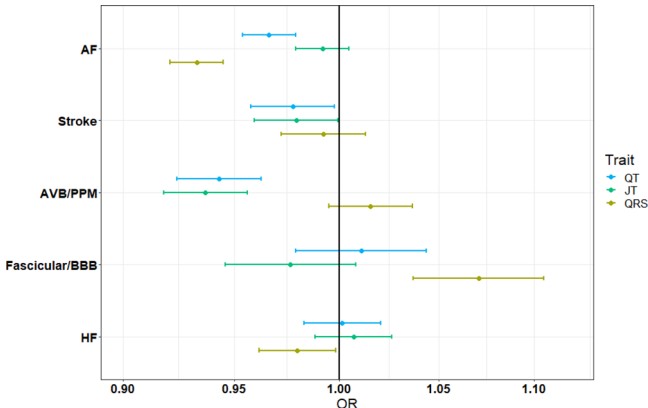

**Fig. 6 | Odds ratios and confidence intervals for ECG PRS with clinical outcomes in UK Biobank.** Data are presented as odds ratios (OR) and 95% confidence intervals (lower 2.5% and upper 97.5%) for association of each ECG (QT – blue, JT – green, QRS – yellow) polygenic risk score (PRS) with prevalent cases in UK Biobank from logistic regression analyses. Associations are reported as risk per standard deviation increase in the PRS and statistical tests were two sided. To adjust for multiple testing, a Bonferroni-corrected threshold ($P < 6.3 \times 10^{-3}$) was used to declare significance. A total of 371,951 individuals of European ancestry were included in this analysis. AF (Atrial Fibrillation), AVB (Atrioventricular block), PPM (Permanent pacemaker), BBB (Bundle branch block), HF (Heart Failure).

understanding of the relationship with common variation in the general population has previously been limited. A recent study suggests the phenotypic effects of rare variants on QT are modulated by polygenic variation in the general population[74]. Our gene-based meta-analyses identified a burden of rare coding variants associated with QT, JT, and QRS, in genes typically linked with inherited channelopathies (*KCNQ1*, *KCNH2*) and cardiomyopathies (*MYH7*, *TNNI3K*). Therefore, our findings support a continuum between the genetic architectures of polygenic traits and disorders that are classically considered monogenic, and highlight the utility of employing a rare-variant gene-based approach in large, unselected populations.

In PRS analyses, we observed decreased risk of AF with increasing QT and QRS PRSs. This is an opposite direction of effect compared with epidemiological studies using directly measured ECG intervals where an increase in QRS or QT is associated with increased risk of AF[75–77]. However, this relationship may be J-shaped, and an increased risk of AF is also observed in patients with short QT syndrome compared to the general population[78,79]. In addition, class-III anti-arrhythmics, used for the chemical cardioversion of AF and maintenance of sinus rhythm, inhibit hERG K + currents that both increase the atrial refractory period (thereby contributing to a protective effect) and prolong the QT interval[80,81]. However, our findings along with the association with conduction disease, may also reflect different biological information captured in the variance explained by the PRS, compared to the directly measured ECG trait; the latter being susceptible to modification by other factors such as coronary artery disease. This may also account for the differences observed when comparing phenotypic and genetic correlations of these traits. Additional research is warranted to investigate these observations. Furthermore, improved risk scores from our study could be used in future work to evaluate the modification of phenotypic expression in families with inherited channelopathies.

While cohorts have extracted ECG parameters using different methods, we believe the large sample sizes and averaging of effect estimates during meta-analysis should limit the impact of any variability on our findings. Of note, we did not observe substantial heterogeneity across results from previously unreported variants. Future GWAS meta-analyses could use the same algorithm across all cohorts to extract ECG phenotypes, but raw digitalized data are not available

for all participants of the current study, so we were unable to do this without substantially reducing the total sample size. Our study includes extensive in-silico follow-up of variants; however, it does not identify causal relationships. Functional follow-up is warranted using the latest advances, including single-cell genomics to further evaluate the relationship of variants with gene-expression[82] and gene-editing tools (e.g. CRISPR) to investigate the effects of coding and regulatory variants on target genes and cellular function in relevant cell types (e.g. human iPSC cardiomyocytes)[83,84].

In summary, by analyzing the largest available sample size to date, we have substantially advanced the delineation of shared and distinct mechanisms influencing ventricular depolarization and repolarization. This work will inform functional follow-up and the prioritization of potential therapeutic targets for arrhythmia.

## Methods
A summary of all input data and tools for each analysis performed in this study is provided in Supplementary Data 24.

### Study cohorts
A total of 35 studies (involving 53 ancestry-specific sub-studies), including members of the Cohorts for Heart and Aging Research in Genomic Epidemiology (CHARGE) consortium[85] contributed to this study (Supplementary Data 1). A maximum total of 252,977 individuals of European (84%), Hispanic (7.7%), African (6.7%), South and South-East Asian (<1%) ancestries were included. All participating institutions approved this project and informed consent was obtained for all individuals at the study level. Cohorts were predominantly population- or community-based with a small number of studies ascertained on a specific case status. Affymetrix or Illumina arrays were typically used for genotyping. Study-specific genotype quality control filters prior to imputation, including call rate, Hardy-Weinberg equilibrium (HWE) *P*-value, and minor allele frequency, are provided in Supplementary Data 2. All GWAS summary data utilized NCBI build 37. Most studies imputed ungenotyped rare and common variants using 1000G reference panels (40/53 sub-studies); the remainder used the Haplotype Reference Consortium (HRC) panel (r1.1 2016), (Supplementary Data 2)[86,87].

Fourteen studies (including 31 sub-studies), imputed genotype data for X chromosome analysis. The pooled multi-ancestry sample size for X chromosome analyses was 86,600 (75,607 European, 7040 Hispanic, 1943 African, 709 South Asian, 590 South East Asian and 711 mixed) for females and 60,343 individuals (52,070 European, 5182 Hispanic, 1518 African, 806 South Asian, 479 South East Asian and 379 mixed) for males.

### Cohort-level single variant association analyses
Single variant genome-wide association studies were performed by each participating cohort for QT, JT, and QRS. For each trait of interest, additive genetic models were implemented for two phenotypes: (1) the raw phenotype (on the millisecond (ms) scale)) and (2) the rank-based inverse normal transformation of each phenotype (on the standard deviation scale due to non-normal distributions of these traits). Per study summary statistics for each ECG measure and covariate are provided in Supplementary Data 3. Individuals were excluded at the study level for: prevalent myocardial infarction or heart failure, pregnancy at the time of recruitment, implantation of a pacemaker or implantable cardiac defibrillator, QRS duration >120 ms, or right or left bundle branch block or atrial fibrillation on ECG. The QRS duration criterion was used as a surrogate marker for bundle branch block and interventricular conduction delay that was not identified during ECG analysis. Additionally, if the data were available, individuals using digitalis, class I or III anti-arrhythmics or QT prolonging medication were excluded. These exclusions were chosen to reduce the risk of confounding in our analyses of ECG parameters, where the bulk of the

power comes from normal variation of QT, JT, and QRS. This will have reduced the total sample size available, however the genetic contribution to ECG interval variation in these disease states could differ, warranting separate investigations. Cohorts including related individuals used appropriate software to account for this e.g BOLT linear mixed model software (BOLT-LMM)[88] (which fits a linear mixed model on hard-called genotyped single nucleotide polymorphisms (SNPs)) or other software incorporating a kinship matrix or pedigree[89–91]. An imputation quality cut-off of Rsq >0.3 (or similar in IMPUTE) was applied in all cohorts to ensure high quality variants were included in the meta-analysis.

Covariates were included in the GWAS model and chosen for their known association with each ECG measure. These included age (years), sex (except in sex-stratified X chromosome analyses), RR interval (ms), height and body-mass index (BMI, kg/m$^2$). Genetic principal components (PCs) were included to account for cryptic population stratification except in cohorts with pedigree data available or when analyses were performed using linear mixed models. As there may be ancestral differences in ECG measure, cohorts comprised of multiple ancestries performed separate analyses for each ancestry to control for underlying population stratification. Separate summary statistics for each ancestry were submitted for central analysis, and for secondary ancestry-specific meta-analyses. Additional cohort-specific covariates were included when deemed appropriate locally, for example, recruitment site or genotyping array. Autosome and X chromosome analyses were performed separately. For X chromosome analyses, male genotypes were coded as 0 or 2 and sex-stratified analyses were performed to account for random X chromosome inactivation in females. Pseudoautosomal regions were excluded from the analyses due to the high risk of genotyping errors in these regions.

## Cohort level generation of covariance matrices for gene-based testing

To test for associations due to a burden of rare (MAF ≤ 0.01) variants with functional consequences within protein-coding genes, additional analyses were performed by participating cohorts using Rare variant test (Rvtests, version 2.0.6)[92]. Rvtests generates summary score statistics per variant using a separate matrix file containing the covariances between markers in a specified sliding window. To avoid associations driven by extreme outliers, these analyses were performed using only rank-based inverse normal transformed values for QT, JT, and QRS. To reduce computational demand, only variants with a MAF < 1% and Rsq >0.3 were included and LD windows of 500 kb were specified for construction of the genotype-covariance matrices. Rvtests uses a genomic kinship matrix to account for relatedness in each cohort and PCs were again included to adjust for residual population stratification. Analyses were only performed using individuals of European ancestry due to the potential of population differences in allele frequencies in rare variants. Only autosomes were included in these analyses.

## Central quality control of study-level data

Quality control of all study-level GWAS summary statistics was performed centrally using the EasyQC R package (version 9.2)[93]. In brief, variants were aligned to either the 1000 G or HRC reference panel and allele frequency plots for each study were compared against the reference. Quantile – quantile (QQ) and P-value – Z-score statistics (P-Z) plots were visually inspected. Variants with invalid beta estimates, standard errors or P-values were removed. Per-study summary statistics were generated including standard error and beta estimate ranges. Genomic-control inflation factors (lambdas) were calculated to identify systematic inflation of test statistics, which can result from a variety of factors, including population stratification, and lead to a large number of true positive findings[94].

## GWAS Meta-Analysis

The meta-analysis workflow is summarized in Fig. 2. The primary analyses were pre-specified to be the multi-ancestry rank-based inverse normal transformed meta-analysis for each ECG trait, to avoid unstable normal approximation test statistics for low-frequency variants or outlier trait values. Ancestry-specific secondary meta-analyses were also performed for European, African, and Hispanic ancestries. Due to the lack of suitably sized replication datasets, we undertook a one-stage, single-discovery design. Additionally, to enable estimation of clinically recognizable effect sizes (inverse normal transformation produces results on a standard deviation scale), a meta-analysis for each trait and ancestry was also performed using the raw phenotype on the millisecond (ms) scale. All meta-analyses were conducted using an inverse variance-weighted, fixed effects model using METAL (version released 2011-03-25) and performed independently across two sites and checked for consistency[95]. Genomic control was applied during meta-analysis to studies in which the inflation factor (λ) was > 1.0. Summary genome-wide association ("Manhattan") plots, QQ plots and lambdas for the entire meta-analysis were produced for each trait using qqman R package (v0.1.8). For all subsequent analyses, only variants present in >50% of the total sample size of the meta-analysis were included. Genome-wide significance (GWS) was defined as $P \leq 5 \times 10^{-08}$. The 1000 Genome reference panel was used for calculating correlations between variants in downstream analyses, including all individuals for multi-ancestry summary statistics and individuals from relevant populations for European, African and Hispanic meta-analyses. Where pre-computed LD scores were required, or correlations calculated within tools that did not permit modification of individuals included in the reference panel, the European ancestry meta-analysis was used in place of the multi-ancestry recognizing a substantial proportion of the multi-ancestry meta-analysis included individuals of European descent. When this is the case, it is explicitly stated in the methodology and results.

## Definition of known and novel loci

To identify novel associations in our results, we first defined boundaries for previously published loci (Supplementary Data 4) using the following process in PLINK(v1.9)[96]. Reported genome-wide significant ($P \leq 5 \times 10^{-08}$) lead variants from each published GWAS were extracted and correlations calculated using the 1000 Genome phase 3 reference panel (Nov 2014)[86] in a 4 Mb region centered on each variant. The locus start was defined as minus 50 kb from the most upstream variant that was in $r^2 > 0.1$ with the lead variant and the locus end as plus 50 kb from the most downstream variant which was in $r^2 > 0.1$. Overlapping boundaries were subsequently merged. This window was declared the genomic boundary for the locus or a minimum physical distance window of ±500 kb around the reported variant – whichever was larger. For the purpose of these analyses, as QT and JT phenotypes are highly correlated phenotypes (Fig. 1), previously reported JT and QT loci were pooled. A separate list was compiled for previously described QRS duration loci. Novel loci in our meta-analyses were identified by applying the same approach to variants meeting the GWS threshold in our results. Variants within loci boundaries not overlapping with known loci were declared novel associations. Subsequently, to evaluate for evidence of heterogeneity at each locus, a forest plot was produced for the lead variant using the R-package Metaviz (version 0.3.0) and manually inspected along with the I$^2$ heterogeneity index for that variant. Finally, Locus-Zoom plots were generated for each locus to visually inspect correlations (r$^2$) between lead variants and surrounding markers and their associated P-values using LocusZoom (v0.12)[97].

## Conditional and heritability analyses

We sought to determine whether any variants at a given locus were conditionally independent (i.e. independent signals of association).

Conditional analyses were performed using Genome-wide Complex Trait Analysis (GCTA, v1.26.0), for loci that achieved genome-wide significance in the European-ancestry analysis with a reference sample of 52,230 individuals of European ancestry from UK Biobank[18]. Related pairs up to the 2nd-degree (kinship coefficient< 0.0884) were excluded. Due to insufficient reference sample size and an inability to effectively reproduce the ancestral mix, conditional analyses were not performed for other ancestries or the multi-ancestry meta-analyses. Stringent thresholds of $P_{Joint} < 5 \times 10^{-08}$ and minimal correlation ($r^2 < 0.1$) with the lead variant were used to declare a variant "conditionally independent". Using the same dataset, heritability estimates for each trait in European samples were obtained using BOLT-Restricted Maximum Likelihood (REML, v2.3.2), which applies variance components analysis using modeled directly genotyped SNPs to calculate SNP-based heritability[88]. The percent variance explained (PVE) by lead and conditionally independent variants was subsequently calculated (Eq. 1)[98];

$$PVE = \frac{[2*(beta^\wedge 2)*MAF*(1 - MAF)]}{[2*(beta^\wedge 2)*MAF(1 - MAF) + ((se(beta))^\wedge 2)*2*N*MAF*(1 - MAF)]} \quad (1)$$

The total PVE by all lead and conditionally independent variants was calculated as the sum of each variant's PVE. The heritability explained was the total PVE divided by the heritability of the trait.

### LD score regression
To calculate the genetic correlation between ECG traits, LD score regression was performed using LD Score (LDSC) software (v1.0.1)[24]. European meta-analysis summary statistics were filtered to include only variants present in the International HapMap Project (-1.1 million variants), along with pre-computed LD scores using the 1000 G reference panel provided by LDSC[24]. LDSC uses the LD scores as regression weights and subsequently calculates the genetic correlation using intersecting SNPs from each meta-analysis[25].

### Gene-based testing meta-analysis
Gene-based meta-analysis was performed using the R package rareMETALs (v.7.1)[99]. Analyses were restricted to up to 192,780 individuals of European descent (from 37 studies) due to potential differences in the allele frequency of rare (MAF≤ 0.01) variants between populations. QC of study-level data was performed as described above for the single-variant meta-analysis. Variants from all studies, however, were subsequently filtered to only include those predicted by VEP (Ensembl release 99) to have high or moderate impact (and thus be protein-altering). Score and covariance files used as input for gene-based meta-analyses in rareMETALS were generated using Rvtests as described above[92]. Gene-based meta-analysis was subsequently performed for inverse-normal transformed QT ($N = 192,780$), JT ($N = 192,501$) and QRS ($N = 192,495$) using Sequence Kernel Association Testing (SKAT)[27], which considers the joint effects of multiple variants on the phenotype, while taking into account the effect size and direction of effect of each variant. Power under SKAT is maximal for genetic variation in a gene that causally increases and decreases a quantitative trait, but is less powered to detect genetic effects that all influence a quantitative trait in one direction. Gene-based meta-analysis was conducted for 18,751 genes that had more than one rare (MAF ≤ 0.01) variant annotated as high or moderate impact. A gene-based test was considered significant if $P < 2.5 \times 10^{-06}$ (Bonferroni correction for ~20,000 tested genes).

To follow-up on gene associations passing Bonferroni correction, additional conditional analyses were performed for each of the three traits. First, to confirm that the gene-based association is not solely driven by one rare variant, conditional gene-based meta-analyses were repeated while conditioning on the most significant variant in the gene. Genes were considered significant if $P_{conditional} < 0.05/$number of

genes tested for the trait (13 genes for QT, 16 genes for JT and 3 genes for QRS), upon gene-based analysis conditioned on the most significant variant. Second, conditional analyses were restricted to 76,202 individuals from the UK Biobank to ensure a common set of variants were able to be examined. To confirm that the gene-based association was not solely driven by flanking low-frequency and common (MAF > 0.01) QT/JT/QRS variants at the locus in which the gene is located regardless of their annotation with VEP, ($P < 5 \times 10^{-08}$ and $r^2 < 0.1$ or ±500 kb from the lead variant in the respective GWAS meta-analysis) analyses were repeated while conditioning on all independent variants at the locus. Furthermore, additional conditional analyses were conducted for associated genes that reside in the same locus. This was done to ensure that these are independent gene associations in the same locus and not attributable to rare variants in the other gene in the locus.

### Biological annotation of GWAS loci
**Identification of variant consequences.** To identify variants with potential functional consequences, we annotated lead and conditionally independent variants, and their proxies ($r^2 > 0.8$), using VEP (Ensembl release 99)[26] to extract information on the impact of a variant on a transcript or protein including their deleteriousness scores using the Sorting Intolerant From Tolerant algorithm (SIFT, version 5.2.2)[100] and PolyPhen-2 (Version 2.2.2)[101]. Additionally, Combined Annotation Dependent Depletion (CADD, v1.4)[102] and RegulomeDB (v.2.0.3)[103] rank scores for these variants were extracted. CADD scores correlate with pathogenicity of both coding and non-coding variants and rate a variant according to its deleteriousness within the genome[102]. RegulomeDB annotates variants with known and predicted regulatory elements in intergenic regions including regions of Dnase hypersensitivity, transcription factors binding sites, and promotor regions utilizing publicly available datasets[103].

**Association with tissue-specific gene expression.** To evaluate correlations between GWAS variants and tissue-specific gene expression, data from the Genotype-Tissue Expression project (GTEx, v8)[33,104,105] were extracted for tissues relevant to cardiac electrophysiology including cardiac, vascular (coronary artery and aorta) and brain (as autonomic regulation influence ECG traits). First, lead and conditionally independent variants and their proxies ($r^2 > 0.8$), were checked to determine whether they overlapped with the lead variant at an expression quantitative trait locus (eQTL) for each tissue. Additionally, to determine whether the same variant may be causal in both our GWAS meta-analysis and the original eQTL study, colocalization analyses were performed using the R package COLOC (v5.1.0), which uses Bayesian methods to determine the correlation between variants from the two datasets[106]. A posterior probability of >75% was used to determine significance.

**Tissue- and cell-type specific regulatory elements.** To identify tissue-specific enrichment of variants in DnaseI hypersensitivity sites, we used GWAS Analysis of Regulatory and Functional Information Enrichment with LD correction (GARFIELD, v2)[37]. GARFIELD performs greedy pruning of GWAS SNPs ($r^2 > 0.1$) and annotates them based on overlapping functional information to assess the enrichment of association signals with features extracted from the ENCODE, GENCODE and Roadmap Epigenomics projects[37]. Odds ratios are quantified and assessed using a generalized linear model framework while matching for MAF, distance to the nearest transcription start site, and number of proxies ($r^2 > 0.8$).

We sought to identify potential target genes of regulatory variants using long-range chromatin interaction (Hi-C) data analyzed using FUMA GWAS (Functional Mapping and Annotation of Genome-Wide Association Studies) software (v1.3.6)[107]. Within FUMA, pre-processed significant loops computed by Fit-Hi-C pipelines filtered at an FDR <

0.05 and overlap with lead and conditionally independent variants and their proxies ($r^2 > 0.8$) was identified[34]. Additionally, we utilized recently published promoter capture Hi-C data which uses loops called from Knight-Ruiz normalized 5 kb, 10 kb and 25 kb resolution data[35]. Promoter interactions in left and right ventricular tissue for potential regulatory variants were extracted and variants with the highest regulatory potential were determined using a RegulomeDB score cut-off of ≤3b[103].

To identify cardiac cell-type specific functional effects of non-coding variants, we integrated GWAS variants with cell-type chromatin marks from single nucleus Assay for Transposase-Accessible Chromatin using sequencing (snATAC-seq) data[36]. These data contain open/accessible chromatin information for nine cell types obtained from the heart, including atrial and ventricular cardiomyocyte, smooth muscle, endothelial, adipocyte, macrophage, fibroblast, lymphocyte and nervous cells. Haplotype blocks were created for each lead and conditionally independent variant including variants with $r^2 > 0.1$ within a 2 Mb radius. Subsequently, using a SNP enrichment method, CHEERS (Chromatin Element Enrichment Ranking by Specificity, v2019)[108], the peaks with the lowest 10th percentile of total read counts from the snATAC-seq data were removed, the peak counts were subsequently quantile normalized, and the Euclidean distance calculated[108]. A one-sided P-value for enrichment of variants in estimated haplotype blocks within cell-type specific ATAC-seq peaks was calculated and a Bonferroni-corrected threshold used to declare significance (0.05/number of cell types).

**Candidate gene prioritization and pathway enrichment.** To prioritize candidate genes at each locus, we used Data-driven Expression-Prioritized Integration for Complex Traits (DEPICT, v3). DEPICT prioritizes the most likely causal genes at associated loci according to common functional pathways using reconstituted gene sets containing a membership probability for each gene in the genome[38]. Additionally, it highlights enriched pathways and tissues/cell types where genes from associated loci are highly expressed. DEPICT uses a clumping method ($r^2 = 0.1$, window size = 250 kb, $P = 5 \times 10^{-08}$) to identify uncorrelated variants from each meta-analysis using 1000 G reference data after excluding the major histocompatibility complex region on chromosome 6. Gene-set enrichment analysis was conducted based on 14,461 predefined reconstituted gene sets from various databases and data types, including Gene Ontology (GO), Kyoto Encyclopedia of Genes and Genomes (KEGG), REACTOME, phenotypic gene sets derived from the Mouse genetics initiative, and molecular pathways derived from protein–protein interactions. Finally, tissue and cell type enrichment analyses were performed based on expression information in any of the 209 Medical Subject Heading (MeSH) annotations for the 37,427 human Affymetrix HGU133a2.0 platform microarray probes. Cytoscape (version 3.8.2, https://cytoscape.org/)[109] was used to visualize significantly enriched (FDR < 0.01) DEPICT GO biological processes for each ECG trait. Processes were connected by overlap of significantly enriched genes (minimum number of 25% of member genes). Orphan pathways or those with less than three edges were excluded. Each module was labeled with a common theme to represent the group of biological processes.

DEPICT requires at least ten genome-wide significant loci to be able to perform the analysis. Therefore, for African and Hispanic ancestries, candidate genes were identified using g:Profiler[110], a functional enrichment tool for the annotation of a list of genes. It also enables mapping of variants to gene names, where they overlap with at least one protein coding Ensembl gene with annotation of predicted variant effects.

In addition to DEPICT gene prioritization results, the list for each locus was supplemented with candidate genes highlighted by these bioinformatic analyses. A literature review was performed which also included a look up of genes using the Online Mendelian Inheritance in Man (OMIM, https://www.omim.org/) and Mouse Genome Informatics (MGI, http://www.informatics.jax.org/) databases.

**Druggability analyses.** To identify potential novel drug targets from our GWAS findings, we interrogated a previously published druggable gene set database developed by Finan et al. which includes detailed information on methods used to assemble and annotate the dataset[40]. In brief, this reference set contains predominantly protein-coding genes (as annotated from the Ensembl v.73). Genes were subsequently assembled into three tiers: Tier 1 includes targets of approved drugs and drugs in clinical development including targets of small molecule and biotherapeutic drugs. Tier 2 incorporates proteins closely related to drug targets or associated with drug-like compounds. Genes where one or more Ensembl peptide sequence shared ≥50% identity (over ≥75% of the sequence) with an approved drug target were included. Tier 3 incorporated extracellular proteins and members of key drug-target families (including G protein-coupled receptors, kinases, ion channels, nuclear hormone receptors, and phosphodiesterases). Using the list of "most likely candidate genes" at unreported GWAS loci and previously reported candidate genes for QT, JT and QRS, a look up was performed in the database. Genes which are existing drug targets for anti-arrhythmic drugs (annotated using KEGG drug (https://www.genome.jp/)) were excluded. As genes which were significant findings in cardiac tissue-specific eQTL co-localization and Hi-C analyses may be favored for prioritization, a look up was performed in our data to highlight these.

**Association between genetically determined QT, JT, and QRS and relevant cardiovascular diseases.** To determine relationships of genetically pre-determined QT JT QRS with cardiovascular diseases PRSs were constructed using the lead variants from the European-ancestry meta-analyses and tested for association with prevalent cases of atrial fibrillation, stroke, coronary artery disease, heart failure, non-ischaemic cardiomyopathy, conduction disease ("AVB or PPM implantation" and "fascicular block/BBB") and ventricular arrhythmia in the UK Biobank. Secondary analyses were also performed to test for association with subgroups including myocardial infarction and stroke. Outcomes were identified using self-reported data, operation codes, ICD-9/ICD-10 codes from hospital episodes statistics and mortality registry (Supplementary Data 25). Analyses were performed in individuals of European-ancestry without ECGs and therefore not included in the GWAS meta-analysis and related pairs up to the 2nd-degree (kinship coefficient <0.0884) were excluded. To take advantage of genotype probability data in BGEN format, PRSice-2 (v2.3.3)[111] was used to calculate the PRS. The PRS was calculated by summing the dosage of the ECG trait prolonging allele, weighted by the effect size from the corresponding GWAS. Associations with prevalent cases were identified using logistic regression including covariates age, sex, 10 PCs and genotype array. A Bonferroni-corrected threshold of 0.05/number of outcomes tested ($0.05/7 = 7.1 \times 10^{-3}$) was used to determine significant associations. P-values < 0.05 but greater than $7.1 \times 10^{-3}$ were considered as suggestive associations.

To determine whether genetically determined QT, JT, and QRS are associated with SCD, PRS were constructed for each trait and tested in four cohorts: the Atherosclerosis Risk in Communities study (ARIC)[112,113], the Cardiac Arrest Blood study (CABS)[114], the Finnish Genetic Study for Arrhythmic Events (FinGesture) and Northern Finland Birth Cohort of 1966 (NFBC1966)[115]. The latter three are independent cohorts as they were not included in the GWAS meta-analysis. While ARIC was included in the GWAS meta-analysis, it contributes a relatively small proportion of the GWAS meta-analysis sample size and thus effects on beta-estimates would be negligible. Individual study information and case definitions used are available in Supplementary Data 22. As FinGesture contains only cases, population controls were taken from NFBC1966[116]. For these analyses, only individuals of

European-ancestry were included, due to low sample sizes of other ancestries. PRSs were constructed by averaging the dosage of each lead variant allele associated with prolongation of the ECG trait being tested, from the European ancestry meta-analysis, weighted by the effect size from the corresponding raw-phenotype meta-analysis. To ensure only high-quality variants were included, each study applied an imputation quality threshold (Rsq > 0.8). The PRS was included in a logistic regression (Cox for ARIC) model along with covariates age (when appropriate), sex, 10 PCs and genotyping array (when appropriate). Sex-stratified analyses were performed to identify sex-specific effects. Per study summary statistics were subsequently meta-analyzed using an inverse-variance weighted fixed effects model with the R package 'Meta' (v.5.5.0)[117].

### Reporting summary

Further information on research design is available in the Nature Research Reporting Summary linked to this article.

## Data availability

Summary statistics from each genome-wide association study meta-analysis will be made available on the NHGRI-EBI Catalog of human genome-wide association studies website, https://www.ebi.ac.uk/gwas/. Electrocardiographic phenotype data derived from UK Biobank digitalized signals, will be returned to the study. The UK Biobank will make these data available to all bona fide researchers for all types of health-related research that is in the public interest, without preferential or exclusive access for any person. All researchers will be subject to the same application process and approval criteria as specified by the UK Biobank. Please see the UK Biobank's website for the detailed access procedure (http://www.ukbiobank.ac.uk/register-apply/). Other datasets used in these analyses are publicly available and can be sourced from: 1000 Genomes reference panel: https://www.internationalgenome.org/category/reference/; Haplotype reference consortium reference panel: http://www.haplotype-reference-consortium.org/; Variant level annotation from Variant Effect Predictor (VEP), Ensembl release 99: https://www.ensembl.org/info/docs/tools/vep/index.html; Variant level Combined Annotation Dependent Depletion scores from Combined Annotation Dependent Depletion (CADD, v1.4): https://cadd.gs.washington.edu/; Variant level tissue-specific gene expression from The GTEx portal (v8): https://gtexportal.org/home/; HiC data from the Functional Mapping and Annotation of Genome-Wide Association Studies (FUMA GWAS, v.1.3.6): https://fuma.ctglab.nl/; DNaseI hypersensivity site enrichment data from GWAS Analysis of Regulatory and Functional Information Enrichment with LD correction (GARFIELD, v2): https://www.ebi.ac.uk/birney-srv/GARFIELD/; Gene-set, biological pathways and tissue expression data from Data-driven Expression-Prioritization Integration for Complex Traits (DEPICT, v3): https://github.com/perslab/depict; Variant level RegulomeDB scores from RegulomeDB (v.2.0.3): https://regulomedb.org/regulome-search/; A compendium of promoter-centered long-range chromatin interactions in the human genome (Jung et al., 2019): https://doi.org/10.1038/s41588-019-0494-8; Cardiac cell type-specific gene regulatory programs and disease risk association (Hocker et al., 2021): https://doi.org/10.1126/sciadv.abf1444; Druggable genome dataset from Finan et al., 2017: DOI: 10.1126/scitranslmed.aag1166; g:Profiler (accessed May 2021): https://biit.cs.ut.ee/gprofiler/gost; Online Mendelian Inheritance in Man database: https://www.omim.org/; Mouse Genome Informatics: http://www.informatics.jax.org/; KEGG drug database: (https://www.genome.jp/).

## Code availability

Codes are available from the original software used for each analysis (see methods and Reporting Summary).

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

## Acknowledgements

All study acknowledgements can be found in Supplementary Note 3, and study funding information in Supplementary Note 4. William J Young was supported by an MRC grant MR/R017468/1. Alvaro Alonso is supported by an NHLBI award K24HL148521. Antonio Luiz P Ribeiro is supported in part by CNPq (310679/2016-8 and 465518/2014-1) and by FAPEMIG (PPM-00428-17 and RED-00081-16). Eduardo Tarazona-Santos, Maria Fernanda Lima-Costa and Thiago P Leal are supported by Brazilian Conselho Nacional de Desenvolvimento Científico e Tecnologico (CNPq). Eduardo Tarazona-Santos and Maria Fernanda Lima-Costa are also supported by the Brazilian Ministry of Health (DECIT/MS, EPIGEN-Brazil Project), Brazilian Ministry of Science and Technology (Financiadora de Estudos e Projetos (FINEP) and Fundação de Amparo a Pesquisa do Estado de Minas Gerais (FAPEMIG). Ruth Loos is supported by the National Institutes of Health (R01DK110113; R01DK107786; R01HL142302; R01DK124097). Michael J Cutler is supported by funding from the Dell Loy Hansen Heart Foundation. M.Benjamin Shoemaker is supported by grant NIHK23HL127704. Rozenn Lemaitre reports funding from grant: AHA 19SFRN34830063. Steven A Lubitz is supported by NIH grant 1R01HL139731 and American Heart Association 18SFRN34250007. Patrick T Ellinor was supported by grants from Bayer AG, the Foundation Leducq (14CVD01), the NIH (1RO1HL092577, R01HL128914, K24HL105780), and the American Heart Association (18SFRN34110082). Lu-Chen Weng was supported by National Institutes of Health (NIH) grant 1R01HL139731 and American Heart Association (AHA) grant 18SFRN34110082. Victor Nauffal is funded by a T32 training grant from the National Institutes of Health (5T32HL007604-35). Nona Sotoodehnia is supported by grants AHA19SFRN348300063, R01HL141989, Medic One Foundation, Laughlin Family. Ulrich Schotten received funding from the Netherlands Heart Foundation (CVON2014-09, RACE V Reappraisal of Atrial Fibrillation: Interaction between hypercoagulability, Electrical remodelling, and Vascular Destabilization in the Progression of AF), the European Union (ITN Network Personalize AF: Personalized Therapies for Atrial Fibrillation: a translational network, grant number 860974; MAESTRIA: Machine Learning Artificial Intelligence Early Detection Stroke Atrial Fibrillation, grant number 965286). Sébastien Thériault is supported by a Junior 1 Clinical Research Scholar award from the Fonds de Recherche du Québec-Santé (FRQS). Jonas L Isaksen is supported by CACHET. Caroline Hayward is supported by an MRC University Unit Programme Grant MC_UU_00007/10 (QTL in Health and Disease). Christy Avery and Antoine Baldassari were supported by NIH grants R01HL142825, and U01HG007416. Dawood Darbar was supported by NIH grants R01HL138737 and T32HL139439. Daniel S Evans is supported by NIH grant U24AG051129. Niels Grarup is supported by the Novo Nordisk Foundation (Grant number NNF18CC0034900). James G.

Wilson is supported by U54GM115428 from the National Institute of General Medical Sciences. Hao Mei is also supported by the CHARGE infrastructure grant (HL105756). Dennis Mook-Kanamori is supported by Dutch Science Organization (ZonMW-VENI Grant 916.14.023). May E Montasser receives funding from Regeneron Pharmaceutical unrelated to this work. James F Wilson and Pau Navarro acknowledge support from the MRC Human Genetics Unit programme grant, "Quantitative traits in health and disease" (U. MC_UU_00007/10). Linda Repetto is funded by a University of Edinburgh studentship. Xia Shen was in receipt of a Starting Grant (2017-02543) from the Swedish Research Council (Vetenskaprådet). Lars Lind is funded by Uppsala University Hospital, Uppsala, Sweden. Niek Verweij was supported by NWO VENI (016.186.125). Jan-Walter Benjamins is funded by the Research Project CVON-AI (2018B017) financed by the PPP Allowance made available by Top Sector Life Sciences & Health to the Dutch Heart Foundation to stimulate public-private partnerships. J.Wouter Jukema is an Established Clinical Investigator of the Netherlands Heart Foundation (grant 2001 D 032). Oliver Hines is supported by a Medical Research Council DTP Studentship. Massimo Mangino is supported by the National Institute for Health Research (NIHR)-funded BioResource, Clinical Research Facility and Biomedical Research Centre based at Guy's and St Thomas' NHS Foundation Trust in partnership with King's College London. Patricia B Munroe, Pier D Lambiase, Andrew Tinker and Michele Orini were supported by Medical Research Council grant MR/N025083/1. Julia Ramirez is funded by European Union's Horizon 2020 research and innovation programme under the Marie Sklodowska-Curie grant agreement No.786833. Pier D Lambiase is supported by UCL/UCLH Biomedicine NIHR, Barts Heart Centre Biomedical Research Centre. Patricia B Munroe, Pier D Lambiase, Andrew Tinker and Helen Warren acknowledge the NIHR Cardiovascular Biomedical Centre at Barts and The London, Queen Mary University of London. Christopher Newton-Cheh acknowledges support from the National Institutes of Health (R01HL143070). The authors also wish to acknowledge the CHARGE infrastructure grant (HL105756).

## Author contributions

Interpreted results, writing, and editing the manuscript: W.J.Y., N.L., A.I., N.S., B.M., C.N.C., P.B.M. Conceptualization of project: N.S., C.N.C., P.B.M. Supervision of project: N.S., B.M., C.N.C., P.B.M. Contributed to GWAS analysis plan: W.J.Y., N.L., N.S., C.N.C., P.B.M. Performed meta-analyses: W.J.Y. and N.L. Performed GCTA, genetic correlations, heritability, variant annotations, GTEx analyses, HiC analyses, gene-set enrichment and pathway analyses, druggability analyses, PRS analyses in UKB: W.J.Y. Performed PRS analyses in SCD cohorts: T.D, J.A.B. Contributed data for PRS SCD analyses: M.-R.J., H.V.H., R.N.M., J.J., P.F., R.L., K.P. Performed gene literature review: W.J.Y., L.F., F.A., R.S., D.R. Visualization of enrichment pathways: S.A.G. Contributed to study-specific GWAS by providing phenotype, genotype and performing data analyses: W.J.Y., A.I., T.D., L.F., J.A.B., R.N., J.-W.B., J.H., L.-P.L., L.Repetto, M.P.C., M.B., S.W., A.R.B., T.M.B., J.P.C., D.S.E., R.F., O.H., J.L.I., H.L., H.M., A.M., M.M.-N., C.N., Y.Q., A.R., C.R., K.R., E.T.-S., S.Thériault, S.V.D., H.R.W., J.Y. S.A., G.A., A.A., L.A., J.C.B., E.B., A.C., E.C., M.C., M.J.C., D.D., A.D.G., A.D.L., J.D., C.E., P.T.E., S.B.F., C.F., M.Gögele, C.G., M.Graff, X.G., T.H., S.R.H., P.L.H., N.H.-K., A.I., R.D.J., M.J., J.J., M.Kavousi, J.A.K., T.P.L., H.J.L., L.L., A.L., S.L., P.M., M.Mangino, T.M., M.Mezzavilla, P.P.M., R.N.M., N.M., M.E.M., A.C.M., M.N., V.N., P.N., K.N., G.Pare, G.Pelliccione, A.P., D.J.P., P.P.P., M.H.P., O.T.R., A.P.R., A.L.P.R., K.M.R., L.Risch, D.S., U.S., C.S., X.S., M.B.S., G.S., M.F.S., E.Z.S., M.S., K.S., K.T., K.D.T., A.T., S.Trompet, A.U., U.V., H.V., M.W., L.-C.W., E.W., J.G.W., C.L.A., D.C., A.C., F.C., M.D., G.G., N.G., C.H., Y.J., J.W.J., S.K., M.Kähönen, J.K.K., C.K., T.L., M.F.L.-C., Y.L., R.J.F.L., S.A.L., D.O.M.-K., A.P.M., J.R.O., M.S.O., M.O., S.P., C.P., A.P., B.M.P., J.I.R., B.S., P.H., C.M.D., N.V., J.F.W., D.E.A., J.R., P.D.L., N.S., P.B.M. All authors read, revised, and approved the manuscript.

## Competing interests

M.J.C. has consulted for Biosense Webster and Janssen Scientific. S.A.L. receives sponsored research support from Bristol Myers Squibb/Pfizer, Bayer AG, Boehringer Ingelheim, Fitbit, and IBM, and has consulted for Bristol Myers Squibb/Pfizer, Bayer AG, and Blackstone Life Sciences. P.T.E. has received grant funding from Bayer AG and served on advisory boards or consulted for Bayer AG, Quest Diagnostics, MyoKardia and Novartis. B.M.P. serves on the Steering Committee of the Yale Open Data Access Project funded by Johnson & Johnson. D.O.M.-K. is a part time research consultant at Metabolon, Inc. D.C has received speaker fees from BMS/Pfizer and consultation fees from Roche Diagnostics. U.S received consultancy fees or honoraria from Università della Svizzera Italiana (USI, Switzerland), Roche Diagnostics (Switzerland), EP Solutions Inc. (Switzerland), Johnson & Johnson Medical Limited, (United Kingdom), Bayer Healthcare (Germany). U.S is co-founder and shareholder of YourRhythmics BV, a spin-off company of the University Maastricht. L.-C.W receives sponsored research support from IBM to the Broad Institute. The remaining authors declare no competing interests.

## Additional information

William J. Young [1,2], Najim Lahrouchi [3,4,5], Aaron Isaacs [6,7], ThuyVy Duong[8], Luisa Foco [9], Farah Ahmed[1], Jennifer A. Brody [10], Reem Salman[1], Raymond Noordam [11], Jan-Walter Benjamins [12], Jeffrey Haessler[13], Leo-Pekka Lyytikäinen [14,15], Linda Repetto[16], Maria Pina Concas [17], Marten E. van den Berg[18], Stefan Weiss [19,20], Antoine R. Baldassari[21], Traci M. Bartz[22], James P. Cook[23], Daniel S. Evans[24], Rebecca Freudling[25,26], Oliver Hines[27,28], Jonas L. Isaksen [29], Honghuang Lin[30,31], Hao Mei[32], Arden Moscati[33], Martina Müller-Nurasyid[26,34,35], Casia Nursyifa[36], Yong Qian[37], Anne Richmond[38], Carolina Roselli [12,39], Kathleen A. Ryan [40,41], Eduardo Tarazona-Santos[42], Sébastien Thériault[43,44], Stefan van Duijvenboden[1,45], Helen R. Warren [1,46], Jie Yao[47], Dania Raza[1,48], Stefanie Aeschbacher[49], Gustav Ahlberg[50,51], Alvaro Alonso [52], Laura Andreasen [50,51], Joshua C. Bis [10], Eric Boerwinkle[53,54], Archie Campbell [55,56,57], Eulalia Catamo[17], Massimiliano Cocca [17], Michael J. Cutler [58], Dawood Darbar [59], Alessandro De Grandi[9], Antonio De Luca[60], Jun Ding[37], Christina Ellervik [61,62,63], Patrick T. Ellinor [39,64], Stephan B. Felix[19,65], Philippe Froguel [66,67,68], Christian Fuchsberger[9,69,70], Martin Gögele [9], Claus Graff[71], Mariaelisa Graff [72], Xiuqing Guo [47,73,74], Torben Hansen [36], Susan R. Heckbert[10,75], Paul L. Huang[76], Heikki V. Huikuri[77], Nina Hutri-Kähönen[78,79,80], M. Arfan Ikram [18], Rebecca D. Jackson[81], Juhani Junttila[77], Maryam Kavousi [18], Jan A. Kors[82], Thiago P. Leal[42,83], Rozenn N. Lemaitre [10], Henry J. Lin[47,73,74], Lars Lind[84], Allan Linneberg [85,86], Simin Liu [87], Peter W. MacFarlane[88], Massimo Mangino [89,90], Thomas Meitinger [26,91,92], Massimo Mezzavilla [17], Pashupati P. Mishra[14,15], Rebecca N. Mitchell[8], Nina Mononen[14,15], May E. Montasser [40,41], Alanna C. Morrison [53], Matthias Nauck [19,93], Victor Nauffal[39,94], Pau Navarro [95], Kjell Nikus[96,97], Guillaume Pare[43], Kristen K. Patton[10], Giulia Pelliccione[17], Alan Pittman[27], David J. Porteous [57,98], Peter P. Pramstaller [9,99], Michael H. Preuss[33], Olli T. Raitakari[100,101,102], Alexander P. Reiner [75,103], Antonio Luiz P. Ribeiro [104,105], Kenneth M. Rice [106], Lorenz Risch [107,108,109], David Schlessinger[110], Ulrich Schotten[6], Claudia Schurmann[33,111,112], Xia Shen [16,113,114], M. Benjamin Shoemaker[115], Gianfranco Sinagra [60], Moritz F. Sinner[25,92], Elsayed Z. Soliman [116], Monika Stoll[7,117,118], Konstantin Strauch[26,34,35], Kirill Tarasov [119], Kent D. Taylor[47,73,74], Andrew Tinker [1,46], Stella Trompet [11,120], André Uitterlinden [121], Uwe Völker[19,20], Henry Völzke[19,122], Melanie Waldenberger[92,123], Lu-Chen Weng [39,124], Eric A. Whitsel[21,125], James G. Wilson[126,127], Christy L. Avery[21], David Conen[43], Adolfo Correa [128], Francesco Cucca[129], Marcus Dörr [19,65], Sina A. Gharib[130], Giorgia Girotto [17,131], Niels Grarup [36], Caroline Hayward [38], Yalda Jamshidi[27], Marjo-Riitta Järvelin [132,133,134,135], J. Wouter Jukema [120,136], Stefan Kääb[25,92], Mika Kähönen[137,138], Jørgen K. Kanters [29], Charles Kooperberg [13], Terho Lehtimäki [14,15], Maria Fernanda Lima-Costa[139], Yongmei Liu[140], Ruth J. F. Loos [33,141], Steven A. Lubitz [39,64], Dennis O. Mook-Kanamori[142,143], Andrew P. Morris [23,144,145], Jeffrey R. O'Connell[40,41], Morten Salling Olesen [51],

**Michele Orini** [2,45], **Sandosh Padmanabhan**[146], **Cristian Pattaro** [9], **Annette Peters** [26,92], **Bruce M. Psaty** [10,75,147], **Jerome I. Rotter**[47,73,148], **Bruno Stricker**[18], **Pim van der Harst** [12,149], **Cornelia M. van Duijn** [150,151], **Niek Verweij** [12], **James F. Wilson** [16,95], **Dan E. Arking**[8], **Julia Ramirez** [1,45], **Pier D. Lambiase**[2,45], **Nona Sotoodehnia**[152,154], **Borbala Mifsud**[1,153,154], **Christopher Newton-Cheh** [4,5,154] ✉ & **Patricia B. Munroe** [1,46,154] ✉

[1]William Harvey Research Institute, Clinical Pharmacology, Queen Mary University of London, London, UK. [2]Barts Heart Centre, St Bartholomew's Hospital, Barts Health NHS trust, London, UK. [3]Amsterdam UMC, University of Amsterdam, Heart Center, Department of Clinical and Experimental Cardiology, Amsterdam Cardiovascular Sciences, Amsterdam, The Netherlands. [4]Program in Medical and Population Genetics, Broad Institute of Harvard and MIT, Cambridge, MA, USA. [5]Cardiovascular Research Center, Center for Genomic Medicine, Massachusetts General Hospital, Boston, MA, USA. [6]Deptartment of Physiology, Cardiovascular Research Institute Maastricht CARIM, Maastricht University, Maastricht, The Netherlands. [7]Maastricht Center for Systems Biology MaCSBio, Maastricht University, Maastricht, The Netherlands. [8]McKusick-Nathans Institute, Department of Genetic Medicine, Johns Hopkins University School of Medicine, Baltimore, MD, USA. [9]Eurac Research, Institute for Biomedicine affiliated with the University of Lübeck, Bolzano, Italy. [10]Cardiovascular Health Research Unit, Department of Medicine, University of Washington, Seattle, WA, USA. [11]Department of Internal Medicine, section of Gerontology and Geriatrics, Leiden University Medical Center, Leiden, The Netherlands. [12]University of Groningen, University Medical Center Groningen, Department of Cardiology, Groningen, The Netherlands. [13]Public Health Sciences Division, Fred Hutchinson Cancer Center, Seattle, WA, USA. [14]Department of Clinical Chemistry, Fimlab Laboratories, Tampere, Finland. [15]Department of Clinical Chemistry, Finnish Cardiovascular Research Center – Tampere, Faculty of Medicine and Health Technology, Tampere University, Tampere, Finland. [16]Centre for Global Health Research, Usher Institute, University of Edinburgh, Edinburgh, Scotland. [17]Institute for Maternal and Child Health – IRCCS "Burlo Garofolo", Trieste, Italy. [18]Department of Epidemiology, Erasmus MC – University Medical Center, Rotterdam, The Netherlands. [19]DZHK German Centre for Cardiovascular Research; partner site Greifswald, Greifswald, Germany. [20]Interfaculty Institute for Genetics and Functional Genomics; Department of Functional Genomics, University Medicine Greifswald, Greifswald, Germany. [21]Department of Epidemiology, Gillings School of Global Public Health, University of North Carolina at Chapel Hill, Chapel Hill, USA. [22]Cardiovascular Health Research Unit, Departments of Biostatistics and Medicine, University of Washington, Seattle, WA, USA. [23]Department of Health Data Science, University of Liverpool, Liverpool, UK. [24]California Pacific Medical Center, Research Institute, San Francisco, CA, USA. [25]Department of Cardiology, University Hospital, LMU Munich, Munich, Germany. [26]Institute of Genetic Epidemiology, Helmholtz Zentrum München - German Research Center for Environmental Health, Neuherberg, Germany. [27]Genetics Research Centre, St George's University of London, London, UK. [28]Department of Medical Statistics, London School of Hygiene and Tropical Medicine, London, UK. [29]Laboratory of Experimental Cardiology, Department of Biomedical Sciences, University of Copenhagen, Copenhagen, Denmark. [30]National Heart Lung and Blood Institute's and Boston University's Framingham Heart Study, Framingham, MA, USA. [31]Section of Computational Biomedicine, Department of Medicine, Boston University School of Medicine, Boston, MA, USA. [32]Department of Data Science, University of Mississippi Medical Center, Jackson, USA. [33]The Charles Bronfman Institute for Personalized Medicine, Icahn School of Medicine at Mount Sinai, New York, NY, USA. [34]IBE, Faculty of Medicine, LMU Munich, Munich, Germany. [35]Institute of Medical Biostatistics, Epidemiology and Informatics IMBEI, University Medical Center, Johannes Gutenberg University, Mainz, Germany. [36]Novo Nordisk Foundation Center for Basic Metabolic Research, Faculty of Health and Medical Sciences, University of Copenhagen, Copenhagen, Denmark. [37]Translational Gerontology Branch, National Institute on Aging, National Institute of Health, Baltimore, US. [38]MRC Human Genetics Unit, Institute of Genetics and Cancer, University of Edinburgh, Western General Hospital, Edinburgh, UK. [39]Cardiovascular Disease Initiative, Broad Institute, Cambridge, MA, USA. [40]Division of Endocrinology, Diabetes, and Nutrition, University of Maryland School of Medicine, Baltimore, MD, USA. [41]Program for Personalized and Genomic Medicine, University of Maryland School of Medicine, Baltimore, MD, USA. [42]Department of Genetics, Ecology and Evolution, Instituto de Ciências Biológicas, Universidade Federal de Minas Gerais, Belo Horizonte/Minas Gerais, Brazil. [43]Population Health Research Institute, McMaster University, Hamilton, Canada. [44]Department of Molecular Biology, Medical Biochemistry and Pathology, Université Laval, Quebec, Canada. [45]Institute of Cardiovascular Sciences, University of College London, London, UK. [46]NIHR Barts Cardiovascular Biomedical Research Centre, Barts and The London School of Medicine and Dentistry, Queen Mary University of London, London, UK. [47]Institute for Translational Genomics and Population Sciences/The Lundquist Institute at Harbor-UCLA Medical Center, Torrance, CA, USA. [48]Brighton and Sussex Medical School, Brighton, UK. [49]Cardiovascular Research Institute Basel, University Hospital Basel, University of Basel, Basel, Switzerland. [50]Laboratory for Molecular Cardiology, The Heart Centre, Department of Cardiology, Copenhagen University Hospital, Rigshospitalet, Copenhagen, Denmark. [51]Department of Biomedical Sciences, University of Copenhagen, Copenhagen, Denmark. [52]Department of Epidemiology, Rollins School of Public Health, Emory University, Atlanta, GA, USA. [53]Human Genetics Center, Department of Epidemiology, Human Genetics, and Environmental Sciences, School of Public Health, The University of Texas Health Science Center at Houston, Houston, TX, USA. [54]Human Genome Sequencing Center, Baylor College of Medicine, Houston, TX, USA. [55]Usher Institute, University of Edinburgh, Nine, Edinburgh Bioquarter, 9 Little France Road, Edinburgh, UK. [56]Health Data Research UK, University of Edinburgh, Nine, Edinburgh Bioquarter, 9 Little France Road, Edinburgh, UK. [57]Centre for Genomic and Experimental Medicine, Institute of Genetics and Cancer, University of Edinburgh, Western General Hospital, Edinburgh, UK. [58]Intermountain Heart Institute, Intermountain Medical Center, Murray, UT, USA. [59]Department of Medicine, University of Illinois at Chicago, Chicago, USA. [60]Cardiothoracovascular Department, ASUGI, University of Trieste, Trieste, Italy. [61]Department of Data and Data Support, Region Zealand, 4180 Sorø, Denmark. [62]Department of Clinical Medicine, Faculty of Health and Medical Sciences, University of Copenhagen, 2100 Copenhagen, Denmark. [63]Department of Laboratory Medicine, Boston Children's Hospital, Harvard Medical School, 300 Longwood Avenue, Boston, MA 02115, USA. [64]Demoulas Center for Cardiac Arrhythmias and Cardiovascular Research Center, Massachusetts General Hospital, Boston, MA, USA. [65]Department of Internal Medicine B - Cardiology, Pneumology, Infectious Diseases, Intensive Care Medicine; University Medicine Greifswald, Greifswald, Germany. [66]Department of Epidemiology and Biostatistics, MRC-PHE Centre for Environment and Health, School of Public Health, Imperial College London, London, UK. [67]University of Lille Nord de France, Lille, France. [68]CNRS UMR8199, Institut Pasteur de Lille, Lille, France. [69]Department of Biostatistics, University of Michigan School of Public Health, Ann Arbor, USA. [70]Center for Statistical Genetics, University of Michigan School of Public Health, Ann Arbor, USA. [71]Department of Health Science and Technology, Aalborg University, Aalborg, Denmark. [72]Department of Epidemiology, University of North Carolina at Chapel Hill, Chapel Hill, NC, USA. [73]Department of Pediatrics/Harbor-UCLA Medical Center, Torrance, CA, USA. [74]Department of Pediatrics/David Geffen School of Medicine at UCLA, Los Angeles, CA, USA. [75]Department of Epidemiology/University of Washington, Seattle, WA, USA. [76]Cardiology Division and Cardiovascular Research Center, Massachusetts General Hospital, Boston, MA 02114, USA. [77]Research Unit of Internal Medicine, Medical Research Center Oulu, University of Oulu and University Hospital of Oulu, Oulu, Finland. [78]Department of Pediatrics, Tampere University Hospital, Tampere, Finland. [79]Department of Pediatrics, Faculty of Medicine and Health Technology, Tampere University, Tampere, Finland. [80]Tampere Centre for Skills Training and Simulation, Faculty of Medicine and Health Technology, Tampere University, Tampere, Finland. [81]Center for Clinical and Translational Science, Ohio State University Medical Center, Columbus, OH, USA. [82]Department of Medical Informatics, Erasmus University Medical Center, Rotterdam, NL, The Netherlands. [83]Lerner

Research Institute, Cleveland Clinic, Cleveland, OH, USA. [84]Deptartment of Medical Sciences, Uppsala University, Uppsala, Sweden. [85]Center for Clinical Research and Prevention, Bispebjerg and Frederiksberg Hospital, Frederiksberg, Denmark. [86]Department of Clinical Medicine, Faculty of Health and Medical Sciences, University of Copenhagen, Copenhagen, Denmark. [87]Center for Global Cardiometabolic Health, Departments of Epidemiology, Medicine and Surgery, Brown University, Providence, USA. [88]Institute of Health and Wellbeing, College of Medical, Veterinary and Life Sciences, University of Glasgow, Glasgow, UK. [89]Department of Twin Research and Genetic Epidemiology, King's College London, London, UK. [90]NIHR Biomedical Research Centre at Guy's and St Thomas' Foundation Trust, London, UK. [91]Institute of Human Genetics, Technical University of Munich, Munich, Germany. [92]DZHK (German Centre for Cardiovascular Research, partner site: Munich Heart Alliance, Munich, Germany. [93]Institute of Clinical Chemistry and Laboratory Medicine, University Medicine Greifswald, Greifswald, Germany. [94]Division of Cardiovascular Medicine, Brigham and Women's Hospital, Boston, MA, USA. [95]MRC Human Genetics Unit, Institute of Genetics and Cancer, University of Edinburgh, Edinburgh, Scotland. [96]Department of Cardiology, Heart Center, Tampere University Hospital, Tampere, Finland. [97]Department of Cardiology, Finnish Cardiovascular Research Center - Tampere, Faculty of Medicine and Health Technology, Tampere University, Tampere, Finland. [98]Centre for Cognitive Ageing and Cognitive Epidemiology, University of Edinburgh, Edinburgh, UK. [99]Department of Neurology, University of Lübeck, Lübeck, Germany. [100]Centre for Population Health Research, University of Turku and Turku University Hospital, Turku, Finland. [101]Research Centre of Applied and Preventive Cardiovascular Medicine, University of Turku, Turku, Finland. [102]Department of Clinical Physiology and Nuclear Medicine, Turku University Hospital, Turku, Finland. [103]Fred Hutchinson Cancer Center, University of Washington, Seattle, WA, USA. [104]Department of Internal Medicine, Faculdade de Medicina, Universidade Federal de Minas Gerais, Brazil, Belo Horizonte, Minas Gerais, Brazil. [105]Cardiology Service and Telehealth Center, Hospital das Clínicas, Universidade Federal de Minas Gerais, Belo Horizonte, Brazil, Belo Horizonte, Minas Gerais, Brazil. [106]Department of Biostatistics, University of Washington, Seattle, WA, USA. [107]Labormedizinisches zentrum Dr. Risch, Vaduz, Liechtenstein. [108]Faculty of Medical Sciences, Private University in the Principality of Liechtenstein, Triesen, Liechtenstein. [109]Center of Laboratory Medicine, University Institute of Clinical Chemistry, University of Bern, Inselspital, Bern, Switzerland. [110]Laboratory of Genetics and Genomics, National Institute on Aging, National Institute of Health, Baltimore, US. [111]Digital Health Center, Hasso Plattner Institute, University of Potsdam, Potsdam, Germany. [112]Hasso Plattner Institute for Digital Health at Mount Sinai, Icahn School of Medicine at Mount Sinai, New York, NY, USA. [113]Department of Medical Epidemiology and Biostatistics, Karolinska Institutet, Stockholm, Sweden. [114]Greater Bay Area Institute of Precision Medicine Guangzhou, Fudan University, Nansha District, Guangzhou, China. [115]Department of Medicine, Division of Cardiovascular Medicine, Arrhythmia Section, Vanderbilt University Medical Center, Nashville, TN, USA. [116]Epidemiological Cardiology Research Center EPICARE, Wake Forest School of Medicine, Winston Salem, USA. [117]Dept. of Biochemistry, Cardiovascular Research Institute Maastricht CARIM, Maastricht University, Maastricht, NL, The Netherlands. [118]Institute of Human Genetics, Genetic Epidemiology, University of Muenster, Muenster, Germany. [119]Laboratory of Cardiovascular Sciences, National Institute on Aging, National Institute of Health, Baltimore, US. [120]Department of Cardiology, Leiden University Medical Center, Leiden, The Netherlands. [121]Internal Medicine, Erasmus MC, Rotterdam, The Netherlands. [122]Institute for Community Medicine, University Medicine Greifswald, Greifswald, Germany. [123]Research Unit Molecular Epidemiology, Institute of Epidemiology, Helmholtz Zentrum München - German Research Center for Environmental Health, Neuherberg, Germany. [124]Cardiovascular Research Center, Massachusetts General Hospital, Boston, MA, USA. [125]Department of Medicine, School of Medicine, University of North Carolina, Chapel Hill, NC 27599, USA. [126]Department of Physiology and Biophysics, University of Mississippi Medical Center, Jackson, USA. [127]Department of Cardiology, Beth Israel Deaconess Medical Center, Boston, USA. [128]Departments of Medicine, Pediatrics and Population Health Science, University of Mississippi Medical Center, Jackson, USA. [129]Institute of Genetic and Biomedical Rsearch, Italian National Research Council, Monserrato, Italy. [130]Center for Lung Biology, Division of Pulmonary, Critical Care and Sleep Medicine, University of Washington, Seattle, WA, USA. [131]Department of Medical Sciences, University of Trieste, Trieste, Italy. [132]Center for Life Course Health Research, Faculty of Medicine, University of Oulu, Oulu, Finland. [133]Unit of Primary Health Care, Oulu University Hospital, Oulu, Finland. [134]Department of Epidemiology and Biostatistics, MRC PHE Centre for Environment and Health, School of Public Health, Imperial College London, London, UK. [135]Department of Life Sciences, College of Health and Life Sciences, Brunel University London, London, UK. [136]Netherlands Heart Institute, Utrecht, The Netherlands. [137]Department of Clinical Physiology, Tampere University Hospital, Tampere, Finland. [138]Department of Clinical Physiology, Finnish Cardiovascular Research Center - Tampere, Faculty of Medicine and Health Technology, Tampere University, Tampere, Finland. [139]Instituto René Rachou, fundação Oswaldo Cruz, Belo Horizonte, Minas Gerais, Brazil. [140]Department of Medicine, Duke University, Durham, NC, USA. [141]The Mindich Child Health and Development Institute, Icahn School of Medicine at Mount Sinai, New York, NY, USA. [142]Department of Clinical Epidemiology, Leiden University Medical Center, Leiden, The Netherlands. [143]Department of Public Health and Primary Care, Leiden University Medical Center, Leiden, The Netherlands. [144]Centre for Genetics and Genomics Versus Arthritis, Centre for Musculoskeletal Research, The University of Manchester, Manchester, UK. [145]Wellcome Centre for Human Genetics, University of Oxford, Oxford, UK. [146]Institute of Cardiovascular and Medical Sciences, University of Glasgow, Glasgow, UK. [147]Health Systems and Population Health, University of Washington, Seattle, WA, USA. [148]Departments of Pediatrics and Human Genetics/David Geffen School of Medicine at UCLA, Los Angeles, CA, USA. [149]Department of Cardiology, Heart and Lung Division, University Medical Center Utrecht, Utrecht, The Netherlands. [150]Nuffield Department of Population Health, University of Oxford, Oxford, UK. [151]Department of Epidemiology, Erasmus MC University Medical Center, Rotterdam, The Netherlands. [152]Cardiovascular Health Research Unit, Division of Cardiology, Department of Medicine, University of Washington, Seattle, WA, USA. [153]Genomics and Translational Biomedicine, College of Health and Life Sciences, Hamad Bin Khalifa University, Doha, Qatar. [154]These authors jointly supervised this work: Nona Sotoodehnia, Borbala Mifsud, Christopher Newton-Cheh, Patricia B. Munroe. ✉e-mail: cnewtoncheh@mgh.harvard.edu; p.b.munroe@qmul.ac.uk

