## [Peer Review File · Nature Communications]

Genetic analyses of the electrocardiographic QT interval and its components identify additional loci and pathwaysREVIEWER COMMENTS

Reviewer #1 (Remarks to the Author):

Young et al. Nature communications 2021 S1

The genetic underpinning of cardiac electrophysiology has taken a large leap in the past 3 decades. This currently includes the ability to perform whole genome analyses to uncover previously unrecognized associations potentially resulting in new pathways that in the future might impact on risk stratification and treatment of individual patients. Importantly, the notion has emerged that there can be large differences between the impact of rare or even common variants with additive or cancellation effects, where there might even be effect differences for different age categories, sex and ethnic origin.

With the current report these colleagues from 154 (!) affiliations share with us their genome-wide ancestry analysis on the QT interval in 252k (!) individuals, including further verification in other cohorts. This is the largest cohort in this field to my knowledge.

The QT interval, a measure of cardiac repolarization on the electrocardiogram, is known for its association with potentially fatal arrhythmia (ventricular fibrillation) when it is excessively prolonged or shortened. This association is most apparent in families with either the rare, Long or, extremely rare, Short QT syndrome. Importantly, within these families where single mutations disrupt normal cardiac electrophysiology, there is a very large variability in their phenotypical expression, from normal to highly malignant with QT related malignant arrhythmias from birth onward. Unmistakable, besides the pathogenic mutation, there thus must be various other mechanisms influencing the QT interval in these patients that finally determines individual phenotypes. Moreover, also outside these inheritable arrhythmia syndromes, in the general population some individuals have a propensity to acquired QT prolongation, intermittently putting them at risk for these same malignant arrhythmias. This might happen in more common circumstances such as cardiac ischemia, bradycardia, electrolyte disturbances, or, e.g., the use of a wide array of cardiac repolarization delaying drugs from psychotropics to antibiotics, that can result in fatal QT prolongation. The current associated concept is that rare or common genetic variant in sync may result in a propensity for no, mild or severe QT prolongation even in the absence of classical pathogenic mutations.

In addition, the QT interval also encompasses cardiac depolarization, depicted on the electrocardiogram as the QRS complex. The QRS complex is by definition included in the QT interval. As a consequence, the QT interval can prolong when cardiac depolarization is impaired, e.g. by conduction disturbances (most notable with 'bundle branch block') but again also in various disease entities such as, again, cardiac ischemia, fibrosis (e.g. cardiomyopathies) and, again, various cardiac depolarization delaying drugs. Impaired cardiac depolarization on itself can already result in malignant arrhythmia, and the combination with impaired cardiac depolarization can result in a further increased for arrhythmia and sudden death.

In this report, these colleagues investigate this concept of multiple genetic modifiers from rare and more common variants that in concert result in phenotypes with less or with more risk for arrhythmia. Moreover, they investigate their association with various mechanistic and clinical entities such as insuline-receptor signaling and atrial fibrillation.

I have the following remarks on this first submission,

Major

- I have several remarks regarding the ECG acquisition and analyses.
- In this project I assume that the ECG parameters were numerically derived from the different cohorts. Although the cohort sizes are enormous, one must acknowledge that cohort specific ECG

analysis methods varied greatly. The consequence being that there is variability in your - most important - primary outcome parameters. I recognize that this is a limitation of almost all large GWAS cohorts but the effect sizes - largely absent in this paper - will certainly overlap with method variability. I would also assume that there could be different effect weights for different cohorts and used models. Please comment

-- The above mentioned issue goes further on the subsequent QT analysis in particular. QT, and in somewhat less extent JT as well, is largely influenced by heart rate, age, sex and ethnicity. Heart rate, for example, may on itself already mirror or explain part of your associations. Please comment

-- The inclusion and exclusion criteria apparently limit excessive depolarization abnormalities (>120ms) but do not exclude repolarization abnormalities, correct? Please comment.

- As mentioned above, in contrast to statistical significance, effect sizes are largely absent. Although the concept is that clinical significant alteration of cardiac electrophysiology can result malignant arrhythmia, and (combined) modifiers of significant alteration are therefore of interest, in most humans/species there is a huge safety factor for both depolarization as well as for repolarization. E.g., for QT a 60-100 millisecond window is easily acceptable for normal to borderline ranges without any clinically significant alteration of risk on individual levels. For QRS this window would be about 40 milliseconds. Of course, polygenic risk scores are able to translate combined effects of genetic modifiers into a certain risk ratio. However, again the potential value remains not very clear. Meanwhile, risk may increase or decrease following subsequent statistical additions of variants with low effect sizes, but probably final risks are also importantly determined by co-existing factors, starting with age, sex and ethnicity, let alone the presence or absence of issues like diabetes etc. Please comment

- Figure 6 is a bit puzzling. Do I understand correctly that there is a negative OR for QRS with (non ischemic?) heart failure, and AF for example? Moreover, the association between QT and stroke and AVB/PM is similarly puzzling. Please comment, also on the inferred underlying mechanisms

- Can you think on a way to illustrate the potential impact of the uncovered variants, possibly in PRS composition, on the broad scale of QRS/QT? I could imagine, for example, a graph with on the X QT / QRS, and on the Y the number of individuals ('normal distribution') and indications on where the various genes/gene combinations exert their effect. What I hope to see is whether there is a particular impact on either side of the clinical spectrum (fast/slow depolarization, repolarization) of specific genes or pathways - indicating ways to use these for further research/development. E.g. when one wants to investigate QT or QRS shortening effects of certain products, or to explain the lack or the presence of severe phenotypes, such investigations could be guided. When there is no such clustering, i.e. even distribution over the phenotypes, the impact of modification of these genes/gene compositions and their pathways is probably much less clear.

Minor

- About the title, I am unsure why you want to focus on QT while you, indeed, investigate both depolarization and repolarization. When you want to mention the ECG parameters, why not use both QRS and QT? In addition, the chronology QT, JT, QRS should, in my view, be reversed throughout.

- 'representing the sum of ventricular depolarization (QRS duration) and repolarization (JT interval)' although this is an often used depiction, the realization is that cardiac repolarization already starts directly after the first depolarized (ventricular) cardiomyocyte, i.e. at the Q. So cardiac repolarization is not entirely depicted by the JT interval. Please rephrase, both in the abstract as well as in the main text. For the analyses you finally group JT and QT, so should these be separated in the text and in the figures?

- Table 2 has an interesting issue where SCN5A is both in QRS and in JT but not in QT. Please comment

- Figure 3, the circular Manhattan is very nice and much appreciated. Can you please think of ways to match the gene text colors with the three circles?

- Figure 5, I do not readily understand the absence vs presence of variant interaction between the 5 biological processes in QT/JT vs QRS.

- 'at 3 loci (NRAP, MYH6, NACA, $P < 1.29 \times 10^{-4}$) but not for' is the 'but' correct here? Would 'and' (no association for) be more appropriate?

- You use the UK biobank at several occasions correct? It seems that there are individuals from this biobank that overlap in several subanalyses. What is the impact hereof?
- 'A Bonferroni threshold (0.05/number of conditions tested) was used to indicate significance ($P < 6.3 \times 10^{-3}$).' here you specifically mention statistical significance whereas in the previous paragraphs you do not. Please comment.
- 'pacemaker implantation (PPM)' you probably mean Permanent PaceMaker?

Reviewer #2 (Remarks to the Author):

In the present manuscript "Genetic analyses of the QT interval and its components in over 250K individuals identifies new loci and pathways affecting ventricular depolarization repolarization" by Young and colleagues, genetic underpinnings of essential electrophysiological process of the human heart were studied.

Remarkably, the authors performed genome-wide ancestry analyses in over 250.000 individuals and found a huge number of new and independent genetic loci for myocardial de- and repolarization. New loci were observed in established pathways for QT- and JT-interval, but also in new genes associated with myocardial energy metabolism. Compared to identified QT and JT loci, connective tissue components and processes for cell growth and extracellular matrix interaction were enriched for alterations in the QRS interval.

By the help of polygenic risk scores (PRs) the authors found also genetic interferences of the identified loci for QRS, QT and JT with further electrical cardiac diseases, such as atrial fibrillation, bradycardia and sudden cardiac death, strongly indicating that identified loci are involved in essential electrical processes of the human heart.

Due the high number of included individuals and the significant clinical relevance of alterations in myocardial de- and repolarization the present study is of high interest for a broad readership. The applied methods and statistical operations seem to be adequately allowing authors data interpretation and data conclusions.

Despite the very interesting topic and the well-written manuscript, there are several shortcomings, which should be addressed during further review process. Please find below my comments:

1. Genome wide association studies (GWAS)

The multi-ancestry GWAS, performed in the present manuscript, deciphered a huge number of novel candidate genes and pathways that might interfere with myocardial de- and repolarization. However, a GWAS does not say anything about causality the specific between the identified allele and the phenotype. Merely, it is an association and more specifically specifically, it is an association only with the polymorphism of a genetic marker and not even directly with a coding allele. A possible causal relationship between genotype and phenotype can only be deciphered by molecular biological and biochemical methods in in vivo models for myocardial electrophysiology, such as zebrafish mammal animal models. Although, the authors try to estimate functional impact of candidate genes by biological annotation of identified GWAS loci (e.g. gene expression levels in cardiac tissue), the presented findings are still associations and not causal relations between the identified genotype and the electrophysiological genotype.

Hence, I suggest to discuss this issue more in detail by giving more information on further scientific efforts that have to be done to identify which candidate gene is truly involved in myocardial electrophysiology.

2. Genotype phenotype relation

It is well known that a huge number of genes coding for myocardial de- and repolarization orchestrate resting and action potential in human cardiomyocytes. As a result of genetic as well as pharmacological studies performed both in humans and in in-vivo models specific function of numerous genes for the phase of myocardial de- and repolarization is well known. Also, it is known, which genes interferes with the specific ECG interval, namely QRS-, QT-, or JT-interval. For example, mutations in SCN5A can cause prolonged QT interval (LQTS3

In the present study, it remains unclear in which direction candidate genes influence QRS, QT or JT. Is there a group of candidate genes associated specifically with QT prolongation or shortening? How was the ECG phenotype analysed? Automatic measurement of QT interval often underestimates length of QT interval, especially if the T wave has an irregular formation, as it can often be found in LQTS. Was U wave in- or excluded from QT interval calculation? Which formula for corrected QT interval was applied? Was baseline heart rate taken into account for choosing QTc formula (QTc calculated by Fridericia in individuals with higher heart rate)?

Next to length of QRS, QT and JT interval, was also formation of QRS, QT or JT taken into account? For example, epsilon waves in case of Brugada syndrome modify significantly QRS interval and is a well-known genetic disease.

After reading the methods section carefully, it seems as electrocardiographic phenotyping only has played a subsidiary role compared to the applied molecular and genetic methods. However, for deciphering the genetic underpinnings of electrophysiological processes in the human heart, sophisticated ECG diagnosis is of the utmost importance.

Reviewer #3 (Remarks to the Author):

This is an interesting study incorporating various genetic analysis techniques using ECG traits to provide new insights into the genetic and biological basis of ventricular depolarization and repolarization. The overall study seems to be well-conducted and methodologically sound, but some revision may be required to improve clarity and readability. Relating to this, I have the following comments:

1. Figure 2 (study design)

- There seems to be a mismatch between the diagram and the text. As far as I am aware, neither the description nor the results from "networks" and "overlap with other traits" analyses are included in the manuscript. On the other hand, there is no mention of rare variant analysis on the diagram.
- The sequential flowchart style seems a bit inaccurate as some analyses can be done in parallel without needing previous steps (e.g. heritability and PRS analyses).
- The label "ARIC, CABS, FinGesture, NFBC1966, SCD meta-analysis" is not clear at first glance without referring to the text

2. Please consider providing a summary of input data and tools that were used in each analysis (e.g. results from the present study or public datasets, genome-wide or independent loci summary statistics, all ancestries or European ancestry only, which reference panels). This can be included in study design (Figure 2) or perhaps in a supplementary table.

3. Can the authors clarify whether the conditional and joint analysis with GCTA was performed: 1) using individual level data in ~50,000 UK Biobank European participants, or 2) using summary statistics from meta-analysis of European participants with reference sample from UK Biobank? The latter makes more sense to me, but this was not clear from the text.

4. Is there any possible explanation why the correlation between QRS and JT is phenotypically almost null (-0.02), but genetically negative (-0.25)? It would be good to include this in the discussion

5. Please clarify the objectives of the PRS analysis as it seems a bit disconnected from the main GWAS meta-analysis.

6. The druggability analysis is a nice touch, but I wonder why targets for existing anti-arrhythmic drugs were excluded from the analysis as these can serve as positive controls to validate the approach if anything comes up.

7. Pathway analysis: Figure 5 (last panel) and Supplementary Figure 7-9 include different traits on different plots, making them difficult to follow. Whenever possible, please include all 3 ECG traits on each plot. Please also clarify how the example GO terms displayed on Figure 5 were selected. Scales on X-axes on Figure 5 and Supplementary Figure 9 also need to be corrected (-log₁₀ P-value should never be negative).

8. I believe results from the heritability analysis can be better utilised in the discussion, for example by comparing results from the present study against the expected population heritability and putting this in the context of genetic architecture to inform future studies (how much variance explained is covered by the present study, do we need a larger sample of common / rare variants association analysis, etc.)

9. Consider removing Figure 1b (phenotype correlation in UK Biobank) as it is not the main focus of the manuscript, and instead just use Supplementary Figure 4 which has both genetic and phenotype correlation. Figure 2 (study design) can then be merged into Figure 1a as an opener to orientate readers. It'd also be good to separate the lower left and upper right triangles on Supplementary Figure 4 and label these with genetic and phenotype correlation to make it clearer without referring to the legend

Manuscript NCOMMS-21-46204-T

Genetic analyses of the QT interval and its components in over 250K individuals identify new loci and pathways affecting ventricular depolarization and repolarization

We thank the reviewers for their comments and suggestions. We have responded in full to each comment and highlighted subsequent revisions to the manuscript in yellow. We believe the revised manuscript has been strengthened, adding clarity to the findings and hope you agree. We have also indicated the location in the manuscript where the corresponding tracked changes can be identified. Edits to supplementary tables are in red font.

We also wish to draw attention to a minor modification to the title where we have changed the word “*identifies*” to “*identify*”. In the original draft the title was:

“Genetic analyses of the QT interval and its components in over 250K individuals identifies new loci and pathways affecting ventricular depolarization and repolarization”.

We have amended to “*Genetic analyses of the QT interval and its components in over 250K individuals identify new loci and pathways affecting ventricular depolarization and repolarization*” which is more grammatically correct.

REVIEWER COMMENTS

Reviewer #1 (Remarks to the Author):

Major

- I have several remarks regarding the ECG acquisition and analyses.

-- In this project I assume that the ECG parameters were numerically derived from the different cohorts. Although the cohort sizes are enormous, one must acknowledge that cohort specific ECG analysis methods varied greatly. The consequence being that there is variability in your - most important - primary outcome parameters. I recognize that this is a limitation of almost all large GWAS cohorts but the effect sizes - largely absent in this paper - will certainly overlap with method variability. I would also assume that there could be different effect weights for different cohorts and used models. Please comment

Authors' reply:

We thank the reviewer for this comment. We agree that there will be variability in the precise algorithm used for extracting ECG variables. ECG analyses were performed by each cohort before participation in this study. While it was not possible to directly control for variability, the summary statistics submitted by participating cohorts for each ECG parameter and covariate, show a broad similarity in their distribution (Supplementary Table 3). To harmonize the GWAS model used by each cohort, we wrote a protocol for each study to follow. This included detailed information on genotype quality control, handling of allelic dosage data, inclusion and exclusion criteria and mandatory covariates. In addition, to avoid associations driven by outliers, our primary meta-analysis for declaring genome-wide independent signals was the rank-based inverse normal transformed phenotype, which will tend to reduce the differences that could exist systematically between cohorts due to using ranks rather than raw values. While ECG method heterogeneity may result in small differences in effect size estimates between cohorts, this will be overcome by the large sample sizes used and the averaging of effect sizes during meta-analysis. For each lead variant reported in our study, we identified no substantial heterogeneity across all previously unreported findings. In our revised manuscript, we have included an additional column in Supplementary Tables 5 (QT), 7 (JT) and 8 (QRS), to provide the heterogeneity I^2 statistics (which were generally low) for each lead variant, as output from the software used to perform the meta-analysis (METAL). We have also included a statement in the discussion to recognize that studies will have used different ECG analysis methods to extract the individual parameters.

Manuscript changes:

Discussion, Page 19:

While cohorts will have extracted ECG parameters using different methods, the large sample sizes and averaging of effect estimates during meta-analysis will limited the impact on our findings and will not influence the identification of positive results. We also observed no substantial heterogeneity across lead variants for all previously unreported findings (Supplementary Tables 5, 7 and 8).

Supplementary Tables 5, 7 and 8:

We have added a column "Het I^2 " containing the heterogeneity I^2 statistic for each lead variant reported for QT, JT and QRS.

-- The above mentioned issue goes further on the subsequent QT analysis in particular. QT, and in somewhat less extent JT as well, is largely influenced by heart rate, age, sex and ethnicity. Heart rate, for example, may on itself already mirror or explain part of your associations. Please comment

Authors' reply:

We agree with the reviewer that the ECG parameters studied are influenced by heart rate, age, sex and ethnicity. To account for this, age, sex, RR interval (the inverse of heart rate), BMI and height,

were included as covariates in the GWAS model. Despite including RR interval as a covariate, we do identify overlap of loci with some previously reported for heart rate. We have indicated these in Supplementary Tables 17-19 (column HR). As our model was adjusted for heart rate, overlap may indicate shared genetic effects between these ECG measures.

In addition, to adjust for underlying population structure (e.g. if systematic differences in ancestry correlated with differences in QT interval), genetic principal components were included (when not accounted for by the GWAS software used). Participating studies were also instructed to perform ancestry-specific GWASs if their cohort included more than one ancestry, and the summary statistics for these were supplied separately. Furthermore, as there may be some ancestral differences in association findings, we performed ancestry-specific meta-analyses as secondary analyses for comparison.

-- The inclusion and exclusion criteria apparently limit excessive depolarization abnormalities (>120ms) but do not exclude repolarization abnormalities, correct? Please comment.

Authors' reply:

Our primary aim was to investigate normal variation affecting ventricular repolarization in a meta-analysis of largely healthy individuals. As the range for each ECG measure followed an expected distribution, the bulk of the power for this study comes from normal variation. In addition, as discussed in our response to the reviewer's first comment, the meta-analysis used for identifying previously unreported findings was the rank-based inverse normal transformed phenotype, which will limit the influence of extreme outliers on our results. We therefore did not exclude individuals with prolonged QT interval for example.

We excluded individuals with a QRS duration greater than 120ms, as a surrogate marker to exclude those with bundle branch block and interventricular conduction delay because such conduction changes are accompanied by lengthening of the heart-rate corrected QT interval. To highlight this, we have amended the methods section to provide the rationale for the use of a QRS >120ms as an exclusion criterion.

Manuscript changes:

Online methods, Pages 21-22:

Individuals were excluded at the study level for: prevalent myocardial infarction or heart failure, pregnancy at the time of recruitment, implantation of a pacemaker or implantable cardiac defibrillator, QRS duration >120ms, or right or left bundle branch block or atrial fibrillation on ECG. **The QRS duration criterion was used as a surrogate marker for bundle branch block and interventricular conduction delay that was not identified during ECG analysis.**

- As mentioned above, in contrast to statistical significance, effect sizes are largely absent. Although the concept is that clinical significant alteration of cardiac electrophysiology can result malignant arrhythmia, and (combined) modifiers of significant alteration are therefore of interest, in most humans/species there is a huge safety factor for both depolarization as well as for repolarization. E.g., for QT a 60-100 millisecond window is easily acceptable for normal to borderline ranges without any clinically significant alteration of risk on individual levels. For QRS this window would be about 40 milliseconds. Of course, polygenic risk scores are able to translate combined effects of genetic modifiers into a certain risk ratio. However, again the potential value remains not very clear. Meanwhile, risk may increase or decrease following subsequent statistical additions of variants with low effect sizes, but probably final risks are also importantly determined by co-existing factors, starting with age, sex and ethnicity, let alone the presence or absence of issues like diabetes etc. Please comment

Authors' reply:

Thank you for raising this important issue. As discussed in response to the previous comment, our primary aim was to study normal variation in a largely healthy population, rather than individuals with extreme values. In addition, we sought to identify robust associations (i.e. statistically significant considering the large number of tests performed) among relatively common variants that would not be expected to have large, individually clinically-significant effects on QT interval with a goal to identify previously unrecognized genetic associations that could ultimately point to novel mechanisms that underlie repolarization. Therefore, in this study we evaluate modest genetic effects individually and then in combination, made possible by the large sample size used. To improve clarity on effect sizes, we have included an additional column to Supplementary Tables 5 (QT), 7 (JT) and 8 (QRS) containing the effect size estimates for each lead variant on the millisecond scale (again, we used the inverse normalized rank-based analysis to declare statistical significance but results from this analysis would be on the standard deviation scale that are less clinically understood).

In this study, we also report the association of each polygenic risk score (PRS) with the directly measured ECG trait in 4214 individuals from UK Biobank. These individuals were chosen as they had ECG data but were not included in the GWAS meta-analysis as the ECGs were not available at the beginning of the study. We agree that co-existing factors will have influence on risk of arrhythmias. To control for some of these, in all PRS analyses, age, sex, BMI, height, RR interval, genotype array used, and 10 genetic principal components, were included as covariates. In the results section, we reported that a standard deviation increase in the PRS was associated with an increase of 6.4ms (5.7–7.1) for QT; 6.4ms (5.7–7.1) for JT; and 2.2ms (1.7–2.7) for QRS. Additional analyses not reported in the submitted draft of the manuscript identified a significant difference (two sample t-test, $P < 2.2 \times 10^{-16}$) when comparing the mean QT/JT interval or QRS duration, for individuals in the top and bottom quintiles of the PRS distribution (16.0ms for QT; 16.0ms for JT and 6.2ms for QRS). To improve clarity and reporting of the results, we have now included this in the results text.

We believe the findings in our study will improve the risk scores for use in future work. For example, to investigate the modification of phenotypic expression in congenital long QT syndrome families. To highlight the potential value of the risk scores reported in the manuscript, we have included a statement in the discussion.

Manuscript changes:

Results, Page 15:

A significant difference in means was observed (two sample t-test, $P < 2.2 \times 10^{-16}$), when comparing individuals in the top and bottom quintiles of the PRS distribution (16.0ms for QT; 16.0ms for JT and 6.2ms for QRS).

Discussion, Page 19:

In PRS analyses, we observed decreased risk of AF with increasing QT and QRS PRSs. This is an opposite direction of effect compared with epidemiological studies using directly measured ECG intervals where an increase in QRS or QT was associated with an increased risk of AF⁷⁵⁻⁷⁷. However, this relationship may be J-shaped as reported in a large study of over 280K individuals, and an increased risk of AF is also observed in patients with short QT syndrome compared to the general population^{78,79}. In addition, class-III anti-arrhythmics, used for the chemical cardioversion of AF and maintenance of sinus rhythm, inhibit hERG K⁺ currents that both increase the atrial refractory period (thereby contributing to a protective effect) and prolong the QT interval^{80,81}. However, our findings along with the association with conduction disease, may also reflect different biological information captured in the variance explained by the PRS, compared to the directly measured ECG trait; the latter being susceptible to modification by other factors such as coronary artery disease. This may also account for the differences observed when comparing phenotypic and genetic correlations of these traits. Additional research is warranted to investigate these observations. Furthermore, improved risk scores from our study could be used in future work to evaluate the modification of phenotypic expression in families with inherited channelopathies.

Supplementary Tables 5, 7 and 8:

We have added a column "Beta (ms)" to include the corresponding effect size estimate for each lead variant from the untransformed GWAS meta-analysis.

- Figure 6 is a bit puzzling. Do I understand correctly that there is a negative OR for QRS with (non ischemic?) heart failure, and AF for example? Moreover, the association between QT and stroke and AVB/PM is similarly puzzling. Please comment, also on the inferred underlying mechanisms

Authors' reply:

Yes, in this study, we report an inverse association between increasing QRS PRS and risk of AF and heart failure. We also observed an inverse association for the QT PRS and AF, stroke and atrioventricular block / permanent pacemaker implantation. We also find these observations are unexpected and we believe them to be of interest to the scientific community. As reported in response to the previous comment, we observed an association between increasing polygenic risk score and each ECG parameter directly measured from the ECG. This provided support that the PRS was correctly constructed and the difference in means when comparing the top and bottom quintiles, is as anticipated with using variants with modest effect sizes.

We discussed potential explanations for the relationship between QT and AF in the discussion, extrapolating from existing knowledge of the influence of class III anti-arrhythmics on the QT interval with protective effects against AF. We also commented on the previously reported J-shaped relationship of the QT interval in a previous large epidemiological study and the association of short QT syndrome with an increased risk of AF compared with the general population. As reported in the results, the proportion of the variance of each ECG trait explained by our findings is approximately 14.6%, 15.9% and 6.3% for QT, JT and QRS respectively. This accounts for 49.8%, 53.9% and 42.0% of the SNP-based heritability of these measures. As presented in the discussion, we suspect that the proportion of the variance of each ECG measure explained by the corresponding PRS, may capture different biological information compared to the directly measured ECG trait. The latter is susceptible to direct modification by other co-existing factors such as ischemic heart disease or hypertension, which may influence the associations reported in large epidemiological studies. This may explain the observed relationship of QRS with heart failure and AF, but we lack the ability to resolve these possibilities from our current study.

We recognize that our study does not identify the underlying mechanisms driving these associations, which is beyond the scope of this current work. We have therefore focused on reporting our observations to the scientific community to facilitate further research into these relationships and at this time we do not wish to overly speculate on the potential drivers of the observed relations beyond the existing text in the manuscript (Discussion, page 19).

- Can you think on a way to illustrate the potential impact of the uncovered variants, possibly in PRS composition, on the broad scale of QRS/QT? I could imagine, for example, a graph with on the X QT / QRS, and on the Y the number of individuals ('normal distribution') and indications on where the various genes/gene combinations exert their effect. What I hope to see is whether there is a particular impact on either side of the clinical spectrum (fast/slow depolarization, repolarization) of specific genes or pathways - indicating ways to use these for further research/development. E.g. when one wants to investigate QT or QRS shortening effects of certain products, or to explain the lack or the presence of severe phenotypes, such investigations could be guided. When there is no such clustering, i.e. even distribution over the phenotypes, the impact of modification of these genes/gene compositions and their pathways is probably much less clear.

Authors' reply:

Thank you for this suggestion. Understanding the relevance of individual variants influencing QT (or QRS) measures to QT variation at different ends of the distribution of QT interval (or QRS duration) would be of substantial interest. Our study of the effect of individual variant effects averaged across all individuals spanning the distribution of values, necessarily combines variant effects occurring in individuals lying at both the upper and lower end of the distribution, since each variant individually has a relatively modest effect on QT interval. Understanding the modification of the genetic effect by an individual's position in the high or low end of the QT distribution is an excellent follow-on study but we lack the ability to make these assessments due to our study design aggregating the results of GWAS analyses performed individually at each cohort, making us unable to reconstruct where any given individual contributing to the study lies in the QT distribution. Our study has identified lead variants at loci that are associated with QT, JT and/or QRS. We have performed extensive in-silico bioinformatic analyses to identify potential candidate genes and plausible affiliated pathways that may explain the associations observed at a variant level. We are, however, unable to determine the exact candidate gene involved, or the potential loss or gain of function effects on these ECG measures. As indicated in response to an earlier comment, while our study has not investigated individuals at the extreme of the ECG parameter distribution, variant effects have a cumulative influence on the directly measured ECG trait and could influence the phenotypic expression in congenital long QT families. A graph could not be constructed with any confidence at this time for the above reasons.

Minor

- About the title, I am unsure why you want to focus on QT while you, indeed, investigate both depolarization and repolarization. When you want to mention the ECG parameters, why not use both QRS and QT? In addition, the chronology QT, JT, QRS should, in my view, be reversed throughout.

Authors' reply:

Thank you for raising this. The primary goal of the study was to investigate variation contributing to the QT interval in a large population study. We focused on QT interval given its primary use in clinical care and greater literature documenting the relationship to arrhythmia risk. However, as the clinically used QT contains both QRS duration and JT interval, we wished to dissect these components and explore their genetic basis along with their relationship with QT. As identified in our study, there are clear discordant directions of effect when comparing JT and QRS, and we hope that by investigating these separately alongside the main QT analysis, we provide greater insight into the underlying genetic contribution to parameters that represent the bulk of ventricular depolarization and repolarization.

Throughout the manuscript, we have presented findings for QT first given this priority, followed by a paragraph for JT and QRS results as a comparison. As our primary aim of the study was to investigate the QT interval, and as QT has a strong genetic correlation with JT interval, we opted for the chronology QT, JT and QRS.

Regarding the title, we note a minor modification is necessary as "Genetic analyses" is plural:

Manuscript changes:

"Genetic analyses of the QT interval and its components in over 250K individuals identify new loci and pathways affecting ventricular depolarization and repolarization"

- 'representing the sum of ventricular depolarization (QRS duration) and repolarization (JT interval)' although this is an often used depiction, the realization is that cardiac repolarization already starts directly after the first depolarized (ventricular) cardiomyocyte, i.e. at the Q. So cardiac repolarization is not entirely depicted by the JT interval. Please rephrase, both in the abstract as well as in the main text. For the analyses you finally group JT and QT, so should these be separated in the text and in the figures?

Authors' reply:

Thank you for making this point. We agree with this point. At a cellular level, cardiac repolarization begins directly after depolarization of the first ventricular cardiomyocyte. At an organ level, the bulk of ventricular depolarization and repolarization occur within QRS duration and JT interval respectively on the surface ECG. We have made modifications to the abstract and main manuscript to improve the accuracy of the text.

We grouped previously reported JT and QT loci due to their high genetic correlation. These were then used to declare previously unreported loci. However, we have otherwise analysed these measures separately throughout. Therefore, and in-line with the original aim of the study, to analyze QT and its individual components, we have presented the JT findings separately. We believe this adds clarity to the underlying genetic basis of each of these measures and highlights key similarities and differences.

Manuscript changes:

Abstract, Page 6:

The QT interval is an electrocardiographic measure representing the sum of ventricular depolarization and repolarization, **estimated by QRS duration and JT interval, respectively. QT interval abnormalities** are associated with potentially fatal ventricular arrhythmia.

Introduction, Page 7:

The electrocardiogram (ECG) is a non-invasive tool that captures cardiac electrical activity¹. The QT interval (QT) represents the sum of ECG measures **that estimate intervals** for ventricular depolarization (QRS duration; QRS) and repolarization (JT interval; JT) at an organ level (Fig. 1).

Introduction, Page 7:

QT and JT phenotypes are highly correlated, whereas QRS has a modest positive and a negligible negative correlation with QT and JT respectively (Fig. 1b)¹⁰. **While at a cellular level, repolarization starts directly after depolarization of the first ventricular cardiomyocyte, at an organ level the majority of ventricular repolarization occurs during the JT interval.**

Discussion, Page 18:

These findings could inform drug development for arrhythmia, as genes or their encoded proteins could be targeted for their specific effects on **predominantly** ventricular depolarization or repolarization.

- Table 2 has an interesting issue where SCN5A is both in QRS and in JT but not in QT. Please comment

Authors' reply:

Thank you for highlighting this. Table 2 shows the findings from the rare variant gene-based meta-analysis. While we identified a statistically significant association of rare variants in aggregate, with QRS and JT, we did not observe this for QT. The P -value for QT was small (2.81×10^{-05}), however this did not pass our Bonferroni adjusted (for $\sim 20,000$ genes tested) threshold to take forward for conditional analysis. This is likely explained by the discordant directions of effect observed for JT and QRS (beta estimates: -0.05 for JT; 0.04 for QRS) that subsequently reduce the strength of the association observed for QT. We have now included this observation in the results.

Manuscript changes:

Results, Page 11:

To investigate whether rare variants (MAF < 0.01) in aggregate modulate ECG traits, we conducted gene-based meta-analyses of rare variants predicted by Variant Effect Predictor (VEP)²⁶ to have high or moderate impact on protein function, using Sequence Kernel Association Testing (SKAT)²⁷. These analyses discovered 13, 16 and 3 genes for QT, JT and QRS respectively ($P < 2.5 \times 10^{-6}$; Bonferroni adjusted for ~20,000 genes). These genes were brought forward for conditional analyses, and 7, 7 and 2 genes remained associated with QT, JT and QRS respectively after conditioning on the rare variant with the lowest P-value at each gene ($P < 0.05/\text{number of genes}$) (Table 2). These results indicate that the gene-based associations were not a consequence of a single variant with a strong effect. We identified an association of rare variants in aggregate at Mendelian long-QT syndrome (LQTS) genes (KCNQ1 [QT and JT], KCNH2 [QT]). SCN5A was associated with JT and QRS, however did not reach the Bonferroni corrected threshold for significance for QT. This could be explained by the discordant directions of effect observed for QRS (Beta [β]: 0.04) and JT (β : -0.05), that subsequently reduce the strength of the association observed for QT (β : -0.03). MYH7 and TNNI3K were also associated with JT. TNNI3K, which was not associated using single variant analysis, encodes a cardiomyocyte-specific kinase previously linked to familial cardiac arrhythmia and dilated cardiomyopathy (OMIM: 613932)^{28,29}.

- Figure 3, the circular Manhattan is very nice and much appreciated. Can you please think of ways to match the gene text colors with the three circles?

Authors' reply:

Thank you for your feedback and for this suggestion. The aim of the circular Manhattan was to highlight the presence of discordant and concordant directions of effect of variants associated with JT and QRS. We therefore chose different colors for the gene text compared with the circular plots, to make it easier to identify loci where there were concordant (green) and discordant (purple) directions of effect. In view of this, our preference would be to keep the current color scheme, although we could revisit this if requested by the editor.

- Figure 5, I do not readily understand the absence vs presence of variant interaction between the 5 biological processes in QT/JT vs QRS.

Authors' reply:

Thank you for your comment and for raising the need to clarify the figure description. Figure 5 was created using the gene-set enrichment findings for the GO biological processes obtained from the DEPICT pathway analyses. For each lead variant at a locus in the meta-analysis, a candidate gene / set of genes, was identified using DEPICT software. Using the same software, GO pathways were identified whether there was significant enrichment ($FDR < 0.01$) of these gene members. In the figure, one ECG panel (QT, JT and QRS), a colored circle represents one of these significantly enriched pathways. These pathways were then linked to another pathway (represented by a light orange line) if they share a gene that is significantly enriched for both pathways. Doing this across all pathways, creates distinct "modules" where pathways are grouped together and common themes can be identified (e.g. "Cardiac & Muscle cell differentiation & development" for all three ECG measures, "Response to Insulin" for QT and JT, Vasculogenesis for QRS). Interestingly, pathways for QRS were more interconnected compared with QT and JT (some links between pathways from different modules). The aim of the figure was to provide an easily accessible visual representation of the pathways identified along with their key biological functions. We have modified the figure description, to provide greater clarity.

Manuscript changes:

Figure 5 caption, Page 43:

The first three panels (QT, JT and QRS) were created using Cytoscape (v3.8.2). Significant GO biological processes (FDR<0.01) from DEPICT pathway analyses (represented as a colored point in the image) were linked together (light orange line) when containing a minimum of 25% overlap of gene members. Orphan pathways or those with less than three edges were excluded. This created discreet "modules" of interlinked pathways, from which common themes could be identified. The final panel shows a bar graph with the most significant GO process members (Y-axis) for JT and QRS from each "common theme", along with their enrichment P-values (X-axis) and color coded by FDR. TGF-beta: Transforming growth factor beta, TRPS/TKS: transmembrane receptor protein serine/threonine kinase.

- 'at 3 loci (NRAP, MYH6, NACA, $P < 1.29 \times 10^{-4}$) but not for' is the 'but' correct here? Would 'and' (no association for) be more appropriate?

Authors' reply:

Thank you for highlighting this and we have modified the text accordingly.

Manuscript changes:

Results, page 8:

There was weaker support for association at 3 loci (NRAP, MYH6, NACA, $P < 1.29 \times 10^{-4}$) and no evidence of support for SUCLA2 ($P > 0.05$) (Supplementary Table 4).

- You use the UK biobank at several occasions correct? It seems that there are individuals from this biobank that overlap in several subanalyses. What is the impact hereof?

Authors' reply:

We do use UK Biobank for multiple analyses as this was the largest dataset included in the study and therefore was recommended for use as a reference sample by some software (see below). The UK Biobank study also contained over 350K individuals not included in the GWAS meta-analysis (as ECG data was not available at the project start) that could be used as an independent cohort for PRS analyses. We provide in the subsequent paragraphs, more detailed information for each analysis.

For GCTA conditional analyses, UK Biobank individuals of European ancestry were included that were also in the GWAS meta-analysis, however this was necessary to perform the analysis. While European-ancestry meta-analysis summary statistics are used to provide the relevant effect sizes, standard errors and P -values, the developers of the GCTA software recommend using the largest participating cohort in the meta-analysis, as the reference sample for calculating LD correlations (<https://yanglab.westlake.edu.cn/software/gcta/#COJO>).

Similarly, to obtain SNP-based heritability estimates, individuals of European ancestry from UK Biobank included in the GWAS meta-analysis were used, as they contribute the largest sample size. However, the percentage variance explained was calculated from the summary statistics of the European-ancestry meta-analysis. This does mean that the portion of heritability explained by lead and conditionally-independent variants is the proportion of SNP-based heritability in these UK Biobank individuals. We have therefore clarified this in the text.

For the additional conditional analyses performed for the gene-based meta-analysis, we used UK Biobank as this was the largest accessible sample size available. The aim of the analyses were to explore the gene-based meta-analysis findings further to determine whether there was a relationship between rare gene-based signals and common or low frequency variants that were independent signals in the single variant GWAS meta-analysis and residing within the same locus as the gene. As the original gene-based meta-analysis and single variant meta-analysis findings were both performed using the full

cohort, it is unlikely that performing these conditional analyses in UK Biobank alone, would substantially affect the results.

For the polygenic risk score analyses, while these were performed in individuals from UK Biobank, there was no overlap with individuals included in the GWAS meta-analysis (they were actively excluded) and therefore they represent an independent cohort.

Manuscript changes:

Results, page 10:

SNP-based heritability estimations in Europeans from UKB for QT, JT and QRS were 29.3%, 29.5% and 15.0% (standard error [SE]: 1%) respectively. The percentage of overall variance explained by all lead and conditionally independent variants from the European meta-analysis was 14.6%, 15.9% and 6.3%. Therefore, these variants explain 49.8%, 53.9% and 42.0% of the SNP-based heritability of QT, JT and QRS **in the UKB individuals included in the heritability estimations.**

- 'A Bonferroni threshold (0.05/number of conditions tested) was used to indicate significance ($P < 6.3 \times 10^{-3}$).' here you specifically mention statistical significance whereas in the previous paragraphs you do not. Please comment.

Authors' reply:

Thank you for this comment. In each paragraph of the results section, we have indicated the thresholds used to declare significance, either by reporting a *P*-value cutoff, posterior probability threshold for analyses using Bayesian statistical methods, or an appropriate false discovery rate. We have not used the words "significant" or "significance" each time to improve readability of the text and due to word count limitations.

- 'pacemaker implantation (PPM)' you probably mean Permanent PaceMaker?

Authors' reply:

Thank you for highlighting this. We have made the necessary corrections.

Manuscript changes:

Results, page 15:

In ~357K unrelated individuals of European ancestry from UKB not included in the GWAS meta-analysis, each PRS was tested for association with prevalent cardiovascular disease cases including atrial fibrillation (AF), "atrioventricular block (AVB) or **permanent** pacemaker implantation (PPM)", "bundle branch block (BBB) or fascicular block", and heart failure (Supplementary Table 21, Supplementary Note 2, Fig. 6).

Figure 6 (caption), page 44:

A total of 371,951 individuals of European ancestry were included in this analysis. AF (Atrial Fibrillation), AVB (Atrioventricular block), PPM (**Permanent** pacemaker), BBB (Bundle branch block), HF (Heart Failure).

Reviewer #2 (Remarks to the Author):

1. Genome wide association studies (GWAS)

The multi-ancestry GWAS, performed in the present manuscript, deciphered a huge number of novel candidate genes and pathways that might interfere with myocardial de- and repolarization. However, a GWAS does not say anything about causality the specific between the identified allele and the phenotype. Merely, it is an association and more specifically, it is an association only with the polymorphism of a genetic marker and not even directly with a coding allele. A possible causal relationship between genotype and phenotype can only be deciphered by molecular biological and biochemical methods in in vivo models for myocardial electrophysiology, such as zebrafish mammal animal models. Although, the authors try to estimate functional impact of candidate genes by biological annotation of identified GWAS loci (e.g. gene expression levels in cardiac tissue), the presented findings are still associations and not causal relations between the identified genotype and the electrophysiological genotype.

Hence, I suggest to discuss this issue more in detail by giving more information on further scientific efforts that have to be done to identify which candidate gene is truly involved in myocardial electrophysiology.

Authors' reply:

Thank you for your comments. We agree that our findings identify associations between variants and ECG parameters and not casual relationships of specific variants or specific genes with the ECG trait. However, we do find association of genetic variation in a specific region with the traits under study and these are not likely to be due to non-genetic effects (e.g. confounding). Confounding for germ-line variation association with biologic traits can only really occur if there is population stratification such that QT interval differs by ancestry and our accounting for ancestry (with ancestry-stratified analyses and use of principal components of ancestry in individual studies) should minimize this risk.

By performing extensive in-silico bioinformatic analyses, including the identification of non-synonymous variants with deleterious effects, expressive quantitative loci colocalization analyses, and Hi-C analyses, we have identified potential candidate genes that may explain associations observed at a variant level. Our findings indicate candidate genes and pathways that could be prioritized for functional follow-up, with the results from the druggability analyses as a possible starting point.

There are multiple potential avenues to further investigate our findings. Single-cell genomics would allow the investigation of relationships between variants identified and subsequent gene-expression at a cellular level and also determine temporal relationships during differentiation (Tanay and Regev, PMID: 28102262). The development of gene-editing tools such as used by the CRISPR system, provide an opportunity to investigate both coding and non-coding regions of the genome through the observation of the effects of potentially causal variants and CREs on target genes and cellular function in relevant cells types such as human iPSC cardiomyocytes (Rao et al, PMID: 33691767) Modelling of LQT syndrome has already been successfully reported using human iPSC derived cardiomyocytes (Sala et al, PMID: 31114684). Our cell-type specific findings already support significant enrichment of variants in atrial and ventricular cardiomyocytes and for JT, adipose tissue. CRISPR based techniques also enable the evaluation of allele substitutions of variants identified in GWAS. For example, non-coding polymorphisms associated with coronary artery disease and stroke in human aortic endothelial cells (Krause et al, PMID: 30429326). We have now included a summary of this in the discussion.

Manuscript changes:

Discussion, Pages 19-20:

Our study includes extensive in-silico follow-up of variants, however it does not identify causal relationships. Functional follow-up is warranted using the latest advances, including single-cell genomics to further evaluate the relationship of variants with gene-expression⁸² and gene-editing tools

(e.g. CRISPR), to investigate the effects of coding and regulatory variants on target genes and cellular function in relevant cell-types (e.g. human iPSC cardiomyocytes)^{83,84}.

2. Genotype phenotype relation

It is well known that a huge number of genes coding for myocardial de- and repolarization orchestrate resting and action potential in human cardiomyocytes. As a result of genetic as well as pharmacological studies performed both in humans and in in-vivo models specific function of numerous genes for the phase of myocardial de- and repolarization is well known. Also, it is known, which genes interferes with the specific ECG interval, namely QRS-, QT-, or JT-interval. For example, mutations in SCN5A can cause prolonged QT interval (LQTS3)

2.1) In the present study, it remains unclear in which direction candidate genes influence QRS, QT or JT. Is there a group of candidate genes associated specifically with QT prolongation or shortening?

Authors' reply:

Thank you for these comments. As addressed in responses to a comment from reviewer 1, while we have identified plausible candidate genes at loci, and have evidence for direction of effect at a variant level, we do not have information regarding the role of specific candidate genes on shortening or prolongation of each ECG measure. Every variant examined has two alleles, one of which is associated with longer QT interval relative to the alternate allele for that variant (sometimes the more common allele, sometimes the less common allele).

2.2) How was the ECG phenotype analysed? Automatic measurement of QT interval often underestimates length of QT interval, especially if the T wave has an irregular formation, as it can often be found in LQTS. Was U wave in- or excluded from QT interval calculation? Which formula for corrected QT interval was applied? Was baseline heart rate taken into account for choosing QTc formula (QTc calculated by Fridericia in individuals with higher heart rate)?

Authors' reply:

Thank you for these comments. ECG phenotypes were derived at a study level using automated methods. This approach has been taken historically in genome-wide association studies due to the very large sample sizes required, that make manual calculation impossible. Despite this, along with our new findings, we identify loci with candidate genes consistently reported in GWAS for these measures and well recognized from other genetic studies in human and in-vivo models, such as *KCNH2*, *KCNE1*, *KCNQ1* and *SCN5A*. Therefore, while there will be some degree of measurement error, it is unlikely to have substantially influenced our findings having averaged the effects across a large number of cohorts and individuals.

It is true that abnormal T-waves and U-waves can make automated measurement of the QT interval challenging, especially in individuals with primary repolarization disorders such as Long QT Syndrome. However, we are focused on community / population-based samples in which LQT individuals are expected to represent only a small proportion (<0.1%). As each heart rate correction formula for the QT interval has its own limitations, we chose to include RR interval as a covariate in the linear GWAS regression model, rather than use a specific correction formula. The linear correction using the RR interval of QT interval to adjust for the heart rate dependence of QT interval has been found to provide more accurate correction than using the square or cube root of RR interval, although these latter measures dominate the clinical QT heart rate correction because of their ease of use. Our approach is the same approach used by other GWAS (Arking *et al*, PMID: 24952745, Méndez-Giráldez *et al*, PMID: 29213071). This same approach was also used to account for heart rate effects on JT and QRS, where no established heart rate correction formula exist, yet an influence on the ECG phenotype is recognized.

2.3) Next to length of QRS, QT and JT interval, was also formation of QRS, QT or JT taken into account? For example, epsilon waves in case of Brugada syndrome modify significantly QRS interval and is a well-known genetic disease.

After reading the methods section carefully, it seems as electrocardiographic phenotyping only has played a subsidiary role compared to the applied molecular and genetic methods. However, for deciphering the genetic underpinnings of electrophysiological processes in the human heart, sophisticated ECG diagnosis is of the utmost importance.

Authors' reply:

In addition to our response to the previous comment, we did not adjust for other ECG measures or findings that may influence the phenotypes included in our study in individuals with specific clinical syndromes such as Brugada syndrome. However, the frequency and role of such electrical findings in large general population cohorts will be limited, due to the small expected numbers of individuals (recognized or unrecognized) with these syndromes.

Reviewer #3 (Remarks to the Author):

1. Figure 2 (study design)

- **There seems to be a mismatch between the diagram and the text. As far as I am aware, neither the description nor the results from “networks” and “overlap with other traits” analyses are included in the manuscript. On the other hand, there is no mention of rare variant analysis on the diagram.**
- **The sequential flowchart style seems a bit inaccurate as some analyses can be done in parallel without needing previous steps (e.g. heritability and PRS analyses).**
- **The label “ARIC, CABS, FinGesture, NFBC1966, SCD meta-analysis” is not clear at first glance without referring to the text**

Authors' reply:

Thank you for highlighting inconsistencies to Figure 2. We have corrected as recommended by removing parts of the diagram that were not included in the final manuscript and added in a panel describing the workflow for the rare variant gene-based meta-analysis. We have reorganized the flowchart to more accurately represent the steps in an order that can be performed. We have also modified the SCD PRS meta-analysis box to improve clarity.

Manuscript changes:

Figure 2, Page 40:

Figure 2: Workflow for single variant analyses of QT, JT and QRS

Workflow for single variant meta-analysis and downstream bioinformatics. VEP (Variant Effect Predictor), CADD (Combined Annotation Dependent Depletion), eQTL (expressive Quantitative Trait Locus), GTEx (Genotype-Tissue Expression project), COLOC (Colocalization), GARFIELD (GWAS Analysis of Regulatory and Functional Information Enrichment with LD correction), DEPICT (Data-driven Expression-Prioritized Integration for Complex Traits), GWAS (Genome-Wide Association Study), EA (European Ancestry), PRS (Polygenic Risk Score), AF (Atrial Fibrillation), CAD (Coronary Artery Disease), CD (Conduction Disease), HF (Heart Failure), NICM (Non-Ischaemic Cardiomyopathy), VA (Ventricular Arrhythmia), SCD (Sudden Cardiac Death)

2. Please consider providing a summary of input data and tools that were used in each analysis (e.g. results from the present study or public datasets, genome-wide or independent loci summary statistics, all ancestries or European ancestry only, which reference panels). This can be included in study design (Figure 2) or perhaps in a supplementary table.

Authors' reply:

Thank you for this suggestion. To avoid over cluttering Figure 2, we have added in a Supplementary Table providing this summary.

Manuscript changes:

Online methods, Page 20:

Online methods

A summary of all input data and tools for each analysis performed in this study, is provided in Supplementary Table 23.

3. Can the authors clarify whether the conditional and joint analysis with GCTA was performed: 1) using individual level data in ~50,000 UK Biobank European participants, or 2) using summary statistics from meta-analysis of European participants with reference sample from UK Biobank? The latter makes more sense to me, but this was not clear from the text.

Authors' reply:

Thank you for raising this. The second statement is true – summary statistics from the meta-analysis of European participants were used with the reference sample from UK Biobank. We have modified the wording to improve the clarity of this sentence in the results and the methods.

Manuscript changes:

Results, Page 8:

To identify additional signals, we performed joint and conditional analyses with Genome-wide Complex Trait Analysis (GCTA)¹⁸ using summary statistics from the European ancestry meta-analysis with the reference sample from UK Biobank (52,230 individuals of European ancestry).

Online methods, Page 25:

We sought to determine whether any variants at a given locus were conditionally independent (i.e. independent signals of association). Conditional analyses were performed using Genome-wide Complex Trait Analysis (GCTA, v1.26.0), for loci that achieved genome-wide significance in the European-ancestry analysis with a reference sample of 52,230 individuals of European ancestry from UK Biobank¹⁸

4. Is there any possible explanation why the correlation between QRS and JT is phenotypically almost null (-0.02), but genetically negative (-0.25)? It would be good to include this in the discussion

Authors' reply:

Thank you for raising this interesting point. A possible explanation is that the directly measured ECG phenotype will be influenced by other co-existing factors such as the ischaemic heart disease and hypertension. This is also the explanation, we believe, for some of the unexpected findings observed in the polygenic risk score analyses, where the directions of effect are different to that observed in large epidemiological studies using the directly measured ECG trait. We have therefore included a sentence to refer to the correlations at the end of the polygenic risk scores paragraph.

Manuscript changes:

Discussion, Page 19:

However, our findings along with the association with conduction disease, may also reflect different biological information captured in the variance explained by the PRS, compared to the directly measured ECG trait; the latter being susceptible to modification by other factors such as coronary artery disease. This may also account for the differences observed when comparing phenotypic and genetic correlations of these traits.

5. Please clarify the objectives of the PRS analysis as it seems a bit disconnected from the main GWAS meta-analysis.

Authors' reply:

As myocardial electrophysiology is influenced by multiple modifiers such as ischemic and non-ischemic heart disease, and capture similar biology that confers risk for conduction disease atrial arrhythmia, we wished to test for association of genetically determined QT, JT and QRS with these relevant cardiovascular diseases. In addition, as prolongation of each ECG parameter is an established risk marker for malignant ventricular and sudden cardiac death, we wished to test for the presence of

an association with our improved risk scores. To clarify the objectives of this section of work, we have made modifications to the relevant section in the results.

Manuscript changes:

Results, Page 15:

PRSs were constructed using European-ancestry lead variants to determine the relationship of genetically determined QT, JT and QRS with the directly measured ECG phenotype, and cardiovascular diseases that may have shared genetic contributions to risk.

Results, Page 15:

As these ECG measures are established risk markers for malignant ventricular arrhythmia and SCD, we also tested each PRS for association with SCD in the Atherosclerosis Risk in Communities (ARIC) study, Cardiac Arrest Blood Study (CABS), Finnish Genetic Study for Arrhythmic Events (FinGesture) and Northern Finland Birth Cohort of 1966 (NFBC1966) cohorts (Supplementary note 3).

6. The druggability analysis is a nice touch, but I wonder why targets for existing anti-arrhythmic drugs were excluded from the analysis as these can serve as positive controls to validate the approach if anything comes up.

Authors' reply:

Thank you for this comment. Existing anti-arrhythmic drug targets such as *KCNH2* and *SCN5A* are well established. We therefore wished to focus the results reporting on potential drug targets that could be prioritized for functional follow up. We have however provided a list of known targets that were excluded, in the caption for supplementary table 20.

7. Pathway analysis: Figure 5 (last panel) and Supplementary Figure 7-9 include different traits on different plots, making them difficult to follow. Whenever possible, please include all 3 ECG traits on each plot. Please also clarify how the example GO terms displayed on Figure 5 were selected. Scales on X-axes on Figure 5 and Supplementary Figure 9 also need to be corrected (-log₁₀ P-value should never be negative).

Author's reply:

Thank you for this suggestion. For figure 5, it will be difficult to readily see differences between QT/JT and QRS if including all three traits in the last panel. Therefore, as QT and JT share all modules, for this panel, we have focused on the differences present for JT and QRS. GO-terms with the smallest enrichment P-values for JT and/or QRS were selected for the plot. We have added this to the figure caption. We have reproduced the plot and corrected the scale on the X-axis.

For Supplementary Figures 7-9, as suggested, we have modified the plots to include all three traits.

Manuscript changes:

Figure 5: Enrichment network visualization of DEPICT GO biological processes

The first three panels (QT, JT and QRS) were created using Cytoscape (v3.8.2). Significant GO biological processes (false discovery rate [FDR]<0.01) from DEPICT pathway analyses (represented as a colored point in the image) were linked together (light orange line) when containing a minimum of 25% overlap of gene members. Orphan pathways or those with less than three edges were excluded. This created discreet “modules” of interlinked linked pathways, from which common themes could be identified. The final panel shows the most significant GO process members (y-axis) for JT and QRS from each “common theme”, along with their enrichment *P*-values (x-axis) and color coded by FDR. TGF-beta: Transforming growth factor beta, TRPS/TKS: transmembrane receptor protein serine/threonine kinase.

Supplementary Figure 7: DEPICT tissue/cell type enrichment analysis

X axis – Nominal P value for enrichment, Y axis – Tissue/cell types. Results are color coded according to the false discovery rate (FDR) value. Detailed results are reported in Supplementary Table 15.

Supplementary Figure 8: Top enriched Gene-Ontology biological processes

Top 20 GO terms (biological processes only) with corresponding $-\log_{10} P$ -values for enrichment (X-axis), for QT (blue), JT (green) and QRS (red). Full results can be found in Supplementary Table 16. A look up of each process was performed in each trait to show corresponding P-values for comparison. Vertical line indicates P-value threshold for declaring significant findings (False discovery rate <0.01). TRPS/TKS: transmembrane receptor protein serine/threonine kinase, RNAP2P: RNA polymerase II promoter.

Supplementary Figure 9: Top enriched Reactome pathways

Top 20 Reactome pathways for QT and JT, and top 11 for QRS (only 11 had a false discovery rate <0.01), with $-\log_{10} P$ -values for enrichment (X-axis), for QT (blue), JT (green) and QRS (red). Full results can be found in Supplementary Table 16. A look up of each pathway was performed in each trait to show corresponding P-values for comparison. Vertical line indicates P-value threshold for declaring significant findings (False discovery rate <0.01). AKT: Protein kinase B, EGFR: Epidermal growth factor receptor, AMPK: AMP-activated protein kinase, FGFR: Fibroblast growth factor receptor, GAB1: GRB2-associated-binding protein 1, KIT: receptor Kit, LKB1: liver kinase B1, MTOR: mammalian target of rapamycin, PI3K: Phosphatidylinositol-3-kinase, PDGF: Platelet-derived growth factor, PKB: Protein kinase B, PP: peroxisome proliferator, PPARa: Peroxisome proliferator-activated receptor alpha, PIP3: Phosphatidylinositol-3,4,5-triphosphate, SCF: Stem cell factor, TGF: Transforming growth factor, TR: Transcriptional regulation.

8. I believe results from the heritability analysis can be better utilised in the discussion, for example by comparing results from the present study against the expected population heritability and putting this in the context of genetic architecture to inform future studies (how much variance explained is covered by the present study, do we need a larger sample of common / rare variants association analysis, etc.)

Authors' reply:

Thank you for this suggestion. As reported in the results section, we identified SNP-based heritability estimates in 52,230 Europeans from UK Biobank for QT, JT and QRS as 29.3%, 29.5% and 15.0%. The percentage of overall variance explained by all lead and conditionally independent variants from the European ancestry meta-analysis was 14.6% (QT), 15.9% (JT) and 6.3% (QRS). Our heritability estimates for QT (and JT as highly correlated with QT) are as anticipated (30-35% for QT [Newton-Cheh *et al*, PMID: 15851319; Nolte *et al*, PMID: 29039294]. Our heritability estimates for QRS are lower than previously reported (23% [95% CI 0.0 – 46] by Nolte *et al*, [PMID: 29039294] and 31.1% [95% CI: 17.3-44.9] by Ritchie *et al*, [PMID: 23463857]). However, the sample sizes in both studies were considerably smaller (N=2,411 and 5,272 respectively) and the confidence intervals very wide. Heritability estimates from twin family studies are higher (36-60% for heart rate corrected QT [Dalageorgou *et al*, PMID: 18031506; Russell *et al*, PMID: 9535482], 34-40% for QRS [Silva *et al*, PMID: 26385552; Busjahn *et al*, PMID: 10377080]), as often is the case in family-based studies as allele-sharing probabilities apply to both common and rare alleles and based on family relationships, while population-based methods (as for our study) may underestimate heritability as the probability of sharing (i.e relatedness) is calculated from predominantly common SNPs. This has previously been studied for ECG traits (Nolte *et al*, PMID: 29039294).

Larger sample sizes using individuals with whole genome sequencing data, will likely increase the percentage variance explained for each ECG trait, however the effect sizes of additional common or low frequency variants identified, will likely be small. Larger studies of rare variation will likely identify a greater proportion of the unexplained heritability, as will studies investigating the interaction of genotype and environmental factors.

While we think this is an interesting discussion, limitation in the word count prevent adding additional text to cover these points. We have however included an additional Supplementary Note to discuss the heritability analyses and comparison with previous studies.

Manuscript changes:

Supplementary Note 4 – Heritability estimates and comparison with previous studies, page 81:

The SNP-based heritability estimates calculated in this study for QT are similar to previously reported values (30-35%). Heritability estimates for JT have not previously been reported in the literature. However, as anticipated due to their high genetic correlation, we obtained similar estimates to QT. Our estimates for QRS are lower than previously reported (23-33%); however, previous calculations were performed on substantially smaller sample sizes resulting in wide confidence intervals. The percentage variance of QT, JT and QRS explained by our findings suggests further studies with larger sample sizes, including individuals with whole genome sequencing data, will likely yield additional loci. However, the effect sizes of additional common variants are likely to be progressively smaller than identified in this study. Larger studies of rare variants may therefore identify a greater proportion of the unexplained heritability, as may gene x environment interaction studies.

9. Consider removing Figure 1b (phenotype correlation in UK Biobank) as it is not the main focus of the manuscript, and instead just use Supplementary Figure 4 which has both genetic and phenotype correlation. Figure 2 (study design) can then be merged into Figure 1a as an opener to orientate readers. It'd also be good to separate the lower left and upper right triangles on Supplementary Figure 4 and label these with genetic and phenotype correlation to make it clearer without referring to the legend

Authors' reply:

Thank you for this suggestion. We have removed Figure 1b from the text in the introduction. Supplementary Figure 4 has subsequently been modified as requested, to separate the lower left and upper right triangles. We prefer to keep Figure 1 and Figure 2 separate as they are referred to in separate sections (Introduction and Methods respectively).

Manuscript changes:

Introduction, page 7:

QT and JT phenotypes are highly correlated, whereas QRS has a modest positive and a negligible negative correlation with QT and JT respectively (Fig. 1b)¹⁰.

Figure 1, page 39:

We have removed Figure 1b and modified the figure caption accordingly.

Figure 1: Annotation of an example ECG signal

QRS duration and the JT interval approximate the time periods for ventricular depolarization and repolarization on the surface ECG. The entire segment from onset of the Q wave to end of the T wave is the QT interval.

Supplementary Figure 4, page 55:

The phenotypic and genetic correlation triangles have been split, color coding swapped (1 = red, -1 = blue) to improve visibility of correlation labels, and colored tiles for genetic correlation changed to circles to highlight that these plots show different data.

Supplementary Figure 4: Genetic Phenotypic and genetic correlations for QT, JT and QRS

Left: Phenotypic correlations (Spearman's rank correlation coefficients [r_s]) were calculated in ~51K UK Biobank individuals of European ancestry. Right: Genetic correlations (r_g) calculated using European ancestry meta-analysis summary statistics with using LDSC regression.

REVIEWERS' COMMENTS

Reviewer #1 (Remarks to the Author):

Young et al. Nature communications. S2

I thank the authors for the rebuttal. I have the following remaining remarks

- regarding the ECG methods. I find the changes made not sufficient. "will limited the impact on our findings and will not influence the identification of positive results" limited should probably be limit. "and will not" is (much) too strong. In addition maybe you intended to use but? And will you provide plans or opportunities for future studies to tackle such issues?

- on QT, heart rate, sex and ethnicity. Please provide (some) additional text on these issues

- about the QRS (and other) exclusions. I would like to see added the why in those exclusions explained in the text, and where this limits the results.

Reviewer #2 (Remarks to the Author):

My concerns regarding ECG phenotyping have not been answered completely. However, i recognize that several major issues have been adressed during the review process leading to an increased qualtiy of the manuscript.

Reviewer #3 (Remarks to the Author):

Thank you for addressing my previous comments.
The revised manuscript reads better and it now looks more suitable for publication in the journal.

Manuscript NCOMMS-21-46204-A

Genetic analyses of the electrocardiographic QT interval and its components identify additional loci and pathways

We thank the reviewers for their comments and the opportunity to further revise the manuscript. We have responded in full to the additional suggestions by reviewer 1 and indicated modifications in the manuscript through tracked changes.

We have also completed all editorial requests and formatting requirements needed for this revision and supply comments in the authors checklist accompanying this submission. These include modification of the main manuscript title to within the 15-word limit.

REVIEWERS' COMMENTS

Reviewer #1 (Remarks to the Author):

Young et al. Nature communications. S2

I thank the authors for the rebuttal. I have the following remaining remarks

- regarding the ECG methods. I find the changes made not sufficient. "will limited the impact on our findings and will not influence the identification of positive results" limited should probably be limit. "and will not" is (much) too strong. In addition maybe you intended to use but? And will you provide plans or opportunities for future studies to tackle such issues?

Authors' reply:

Thank you for identifying the typo. We have revised the sentence in the discussion and added a suggestion to use a single algorithm across all cohorts to harmonize ECG annotation in future studies.

Manuscript changes

Discussion Page 19

"While cohorts have extracted ECG parameters using different methods, we believe the large sample sizes and averaging of effect estimates during meta-analysis should limit the impact of any variability on our findings. Of note, we did not observe substantial heterogeneity across results from previously unreported variants. Future GWAS meta-analyses could use the same algorithm across all cohorts to extract ECG phenotypes, but raw digitalized data are not available for all participants of the current study, so we were unable to do this without substantially reducing the total sample size."

- on QT, heart rate, sex and ethnicity. Please provide (some) additional text on these issues

Authors' reply:

We have now edited text in the methods section to highlight how we addressed these issues.

Manuscript changes

Methods Page 22

"Covariates were included in the GWAS model and chosen for their known association with each ECG measure. These included age (years), sex (except in sex-stratified X chromosome analyses), RR interval (ms), height and body-mass index (BMI, kg/m²). Genetic principal components (PCs) were included to account for cryptic population stratification except in cohorts with pedigree data available or when analyses were performed using linear mixed models. As there may be ancestral differences in ECG measures, cohorts comprised of multiple ancestries performed separate analyses for each ancestry to control for underlying population stratification. Separate summary statistics for each ancestry were submitted for central analysis, and for secondary ancestry-specific meta-analyses."

- about the QRS (and other) exclusions. I would like to see added the why in those exclusions explained in the text, and where this limits the results.

Authors' reply:

We have now edited text in the methods section to highlight why these exclusions have been applied and how they could limit the results.

Manuscript changes

Methods Page 21

"Individuals were excluded at the study level for: prevalent myocardial infarction or heart failure, pregnancy at the time of recruitment, implantation of a pacemaker or implantable cardiac defibrillator, QRS duration >120ms, or right or left bundle branch block or atrial fibrillation on ECG. The QRS duration criterion was used as a surrogate marker for bundle branch block and interventricular conduction delay that was not identified during ECG analysis. Additionally, if the data were available, individuals using digitalis, class I or III anti-arrhythmics or QT prolonging medication were excluded. These exclusions were chosen to reduce the risk of confounding in our analyses of ECG parameters, where the bulk of the power comes from normal variation of QT, JT, and QRS. This will have reduced the total sample size available, however the genetic contribution to ECG interval variation in these disease states could differ, warranting separate investigations."

Reviewer #2 (Remarks to the Author):

My concerns regarding ECG phenotyping have not been answered completely. However, i recognize that several major issues have been addressed during the review process leading to an increased quality of the manuscript.

Authors' reply:

We thank the reviewer for their time in reviewing our revised manuscript.

Reviewer #3 (Remarks to the Author):

Thank you for addressing my previous comments. The revised manuscript reads better and it now looks more suitable for publication in the journal.

Authors' reply:

We thank the reviewer for their time in reviewing our revised manuscript.